

# Two-dimensional Numerical Simulations of Mixing under Ice Keels

Sam De Abreu[1], Rosalie M. Cormier[1], Mikhail G. Schee[1], Varvara E. Zemskova[2], Erica Rosenblum[3,*], and Nicolas Grisouard[1,*]

[1]University of Toronto, Department of Physics, 60 St. George Street, Toronto ON M5S 1A7, Canada
[2]College of Earth, Ocean, and Atmospheric Sciences, Oregon State University, Corvallis, Oregon, USA
[3]Centre for Earth Observation Science, University of Manitoba, Winnipeg, Manitoba, Canada
[*]These authors contributed equally to this work.

**Correspondence:** Sam De Abreu (sam.b.deabreu@gmail.com)

**Abstract.**

Changes in sea ice conditions directly impact the way the wind transfers energy to the Arctic Ocean. The thinning and increasing mobility of sea ice is expected to change the size and speed of ridges on the underside of ice floes, called ice keels, which cause turbulence and impact upper-ocean stratification. However, the effects of changing ice keel characteristics on below-ice mixing are difficult to determine from sparse observations and have not been directly investigated in numerical or laboratory experiments. Here, for the first time, we examine how the size and speed of an ice keel affect the mixing of various upper-ocean stratifications using 16 two-dimensional numerical simulations of a keel moving through a two-layer flow. We find that the irreversible ocean mixing and the characteristic depth over which mixing occurs each vary significantly across a realistic parameter space of keel sizes, keel speeds, and ocean stratifications. Furthermore, we find that mixing does not increase monotonically with ice keel depth and speed, but instead depends on the emergence and propagation of vortices and turbulence. These results suggest that changes to ice keel speed and depth may have a significant impact on below-ice mixing across the Arctic Ocean, and highlight the need for more realistic numerical simulations and observational estimates of ice keel characteristics.

## 1 Introduction

Wind-driven mixing in the Arctic Ocean directly affects the upper-ocean stratification and, therefore, the vertical transfer of heat and the evolution of sea ice (Carmack et al., 2015; Rainville et al., 2011; MacKinnon et al., 2021; Timmermans and Marshall, 2020). Processes that drive mixing typically occur at spatiotemporal scales smaller than a climate model can resolve; hence, ocean mixing has to be parameterized. Despite its importance, the representation of ocean mixing in ice-covered waters is poorly constrained in climate models, contributing to an unrealistic representation of the Arctic halocline in the past several generations of global ice-ocean models and coupled climate models (Holloway et al., 2007; Ilicak et al., 2016; Rosenblum et al.,



2021; Barthélemy et al., 2015; Jin et al., 2012; Nguyen et al., 2009; Sidorenko et al., 2018). This has direct implications for biases in simulated circulations of Pacific and Atlantic Water and ice-ocean heat fluxes (Zhang and Steele, 2007; Lavoie et al., 2021; Liang and Losch, 2018) as well as sea ice retreat (Niederdrenk and Notz, 2018; Rosenblum and Eisenman, 2016, 2017; Notz and SIMIP Community, 2020; Stroeve et al., 2007, 2012; Winton, 2011).

Historically, multi-year sea ice shielded the majority of the underlying Arctic Ocean from the wind, leading to a relatively quiescent Arctic Ocean. However, during the period 1979-2018, the September ice cover receded by 45.2% relative to 1979-1989 (Stroeve and Notz, 2018). The shrinkage of this "sea ice lid" has allowed the wind to interact directly with the ocean, potentially increasing wind-driven momentum transfer into the ocean and increasing vertical ocean mixing (Rainville and Woodgate, 2009; Rainville et al., 2011; Rippeth et al., 2015; Brown et al., 2020).


But what happens below a thick, multi-year sea ice cover that is transitioning to thinner, first-year ice? Like a spoon in a glass of water, sea ice stirs and mixes the ocean differently, depending on its shape and speed. Observations indicate that the shapes and speeds of sea ice have undergone significant changes over the past few decades. The proportion of multi-year to first-year sea ice has declined from 59% in 1984 to 28% in 2018 (Stroeve and Notz, 2018), resulting in a sea ice pack that is on average thinning (Kwok and Rothrock, 2009; Kwok, 2018; Rampal et al., 2011), becoming more mobile with faster drift speeds (Rampal et al., 2009) and shallower ridges (Wadhams, 2012; Hutchings and Faber, 2018; Kwok, 2018), consisting of smaller ice floes (Hutchings and Faber, 2018), and undergoing large regional and temporal variations in roughness (Tsamados et al., 2014).


Using global ice-ocean model simulations, Martin et al. (2016) demonstrated that the shape of sea ice matters as much as its speed to explain the decadal basin-wide changes in wind-driven momentum transfer into the ocean. Specifically, they found that the inclusion of changes to roughness of sea ice, modeled as sea ice keels (deep pressure ridges underneath ice floes) and sails (the keels' above-ice counterparts), caused ocean surface stress to decrease by 3.1% per decade. On the contrary, simulations that ignore changes to sea ice roughness, such as most climate models, simulate an ocean surface stress that increases by 4.6% per decade. This sensitivity to sea ice roughness is qualitatively consistent with observational, numerical, and laboratory studies that indicate that sea ice bottom roughness is important for kinetic energy dissipation, vertical mixing (Fer and Sundfjord, 2007; Fer et al., 2022), internal wave generation (Cole et al., 2018), and ice-ocean drag (Cole et al., 2017; Brenner et al., 2021; Pite et al., 1995; Zu et al., 2021), and has direct implications for ice-ocean turbulent heat fluxes (Randelhoff et al., 2014; Skyllingstad et al., 2003).



Oceanographic studies suggest that ice keels cause significant turbulence that should have a large impact on the summer pycnocline. In situ observations by Fer and Sundfjord (2007) and Fer et al. (2022) report that the presence of keels yields deeper mixing and greater kinetic energy dissipation than would be expected if the ice was flat. Zhang et al. (2022) ran numerical experiments of internal solitary waves (non-linear, non-hydrostatic oscillations of the pycnocline) impinging on keels, reporting that this interaction can result in the creation of secondary waves and turbulence. Using three-dimensional large-eddy simulations inspired by Arctic observations, Skyllingstad et al. (2003) estimated that ice keels could enhance vertical heat fluxes by factors of 3 to 10, but did not distinguish between reversible fluxes (*i.e.*, adiabatic stirring of water parcels) and irreversible fluxes (*i.e.*, diabatic mixing of waters of different densities). This distinction matters for climate models because







the latter has long-term impacts on the stratification of the upper ocean, while the former does not. Here, for the first time, we directly investigate the relative impacts of ice keel size and speed on diabatic turbulent ocean mixing.

Accurate representations of the impact of sea ice roughness on ocean mixing in climate model parameterizations will require

a fundamental understanding of how the variability in ice keels' sizes and shapes affects mixing in the upper ocean. How ice keel-like objects interact with Arctic-like summer mixed layers has, to our knowledge, been the topic of relatively few process studies. In this article, we explore keel-driven mixing considering a range of ice keel depths, ice keel speeds, and summer upper-ocean stratifications that are representative of previous observations in the Arctic using a suite of two-dimensional numerical experiments.

In Section 2, we describe how we adapt several previous approaches to characterize our numerical simulation setup in the context of ice keel stirring and mixing, as well as the governing equations, and our approach to quantifying mixing. In Section 3, we categorize the upstream of each simulation into one of three stirring regimes and the downstream into one of four stirring regimes.

Intuitively, we could expect faster, deeper keels to generate more diapycnal mixing over a larger depth range (hereafter

referred to as the null-hypothesis). However, relatively deep keels can also act to inhibit mixing by blocking flow ahead (upstream) of the keel. Similarly, relatively fast keels can act to inhibit mixing by generating a localized boundary layer, allowing laminar flow above to move past the keel with little interaction between layers of different density or with the keel. We use our stirring regime categorization to interpret variations away from the null-hypothesis and present the magnitude and vertical extent of mixing in Section 4. We further discuss and summarize, putting our results in the context of the Arctic Ocean,

in Sections 5 and 6.

## 2   Methods

### 2.1   Governing Equations

We use an idealized two-layer model to simulate the cross-sectional motion of fluid around an ice keel. Figure 1 shows each element of our setup, which we expand on in the rest of this section. Our equations and code are adapted from Hester et al.

(2021), who simulated a single-layer fluid flow across an iceberg at the surface to examine the influence of its aspect ratio on melt. Our specific adaptations are the use of a two-layer rather than one-layer stratification, adjustment of the shape of the obstacle to simulate an ice keel rather than an iceberg, inclusion of a gradual acceleration of the obstacle, removal of all possible phase changes of water, and removal of temperature dependence from the governing equations. We ignore Coriolis terms because our time and length scales are small (tens of minutes and tens of meters, respectively).




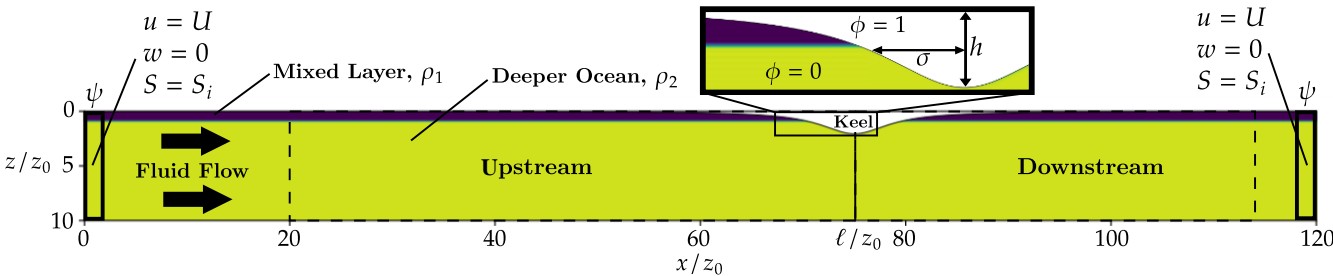

**Figure 1.** A schematic of our setup, which consists of a mixed layer (dark blue) initialized over $z \leq z_0$ with density $\rho_1$, a deeper ocean layer (yellow) with density $\rho_2$, and an ice keel (white) located at $x = \ell$. Dashed lines indicate the upstream region ($20 \leq x/z_0 \leq 75$) and downstream region ($75 \leq x/z_0 \leq 115$), both extending down the entire water column while excluding the keel. Sponge layers ($x/z_0 \leq 2.5$ and $x/z_0 \geq 117.5$), denoted by $\psi$, and their associated governing equations are also indicated.

The resulting governing equations form a set of non-hydrostatic salinity advection-diffusion equations with fluid velocity, pressure, and density fluctuations that satisfy the Boussinesq equations, namely,

$$\frac{\partial u}{\partial t} + \frac{\partial p}{\partial x} - \nu \frac{\partial q}{\partial z} = -wq - \frac{\phi}{\xi}u - \frac{\psi}{\xi}(u - U), \tag{1}$$

$$\frac{\partial w}{\partial t} + \frac{\partial p}{\partial z} + \nu \frac{\partial q}{\partial x} = uq - \frac{\phi}{\xi}w - \frac{\psi}{\xi}w + g\frac{\rho(S) - \rho_0}{\rho_0}, \tag{2}$$

$$\frac{\partial S}{\partial t} - \mu \nabla^2 S = -u\frac{\partial S}{\partial x} - w\frac{\partial S}{\partial z} - \frac{\mu}{1 - \phi + \delta}\left(\nabla S \cdot \nabla \phi\right) - \frac{\phi}{\xi}(S - S_i) - \frac{\psi}{\xi}(S - S_i), \tag{3}$$

$$\frac{\partial u}{\partial x} + \frac{\partial w}{\partial z} = 0, \tag{4}$$

along with the EOS-80 equation of state (Fofonoff and Millard, 1983) that relates the density $\rho$ to the salinity $S$, assuming a constant temperature of $-2°$C. Here, $x$ is the horizontal distance from the left boundary; $z$ is the vertical distance from the surface (that is, it increases positively downward); $u$ and $w$ are the horizontal and vertical velocity components, respectively; $p$ is the scaled pressure fluctuation field; $S_i$ is the initial salinity field (see Eq. 5); $q \equiv \partial_z u - \partial_x w$ is the spanwise vorticity; $U$ is the

keel velocity relative to the fluid; $\nu$ and $\mu$ are the kinematic viscosity and the salt-mass diffusivity, respectively (both of value $2 \times 10^{-3} \, \mathrm{m^2 \, s^{-1}}$); and $\nabla = (\partial_x, \partial_z)$. We choose such large values for $\nu$ and $\mu$ to dissipate eddies smaller than the resolution of the grid, similar to the choice of Zhang et al. (2022). The fields $\phi$ and $\psi$ are masks for the keel and sponge layers (see below), $\xi = 7.1$ ms is the damping and restoring time scale associated with $\phi$ and $\psi$, and $\delta = 5 \times 10^{-3}$ regulates the salinity equation within the keel.

The two-layer fluid has an initial salinity profile $S_i(x, z)$, which consists of an upper layer of salinity $S_1$ and density $\rho_1 = \rho(S_1)$, which we call the "mixed layer," overlying a deeper layer of salinity $S_2$ and density $\rho_2 = \rho(S_2) > \rho_1$, which we call the "deeper ocean". The mixed layer extends to a depth of $z_0$ with a narrow halocline transition into the deeper ocean layer. The



initial salinity profile is

$$
S_i(x,z) = \begin{cases} S_1 \phi(x,z) & \text{if } z \leq H(x), \\ S_1 + \frac{\Delta S}{2}\left[1 - \tanh\left(\frac{z_0-z}{b}\right)\right] & \text{otherwise,} \end{cases} \tag{5}
$$

where $H(x)$ is the location of the keel boundary, which we define below, $\Delta S = S_2 - S_1$, and $b = 0.1$ m.

Inside the keel, the salinity is maintained at $S_1$, the initial salinity of the mixed layer. The smooth phase field $\phi(x,z)$ marks the location of the keel. That is, it satisfies $\phi = 1$ inside the keel and $\phi = 0$ outside, with a sigmoidal transition over a distance $\varepsilon$ across the boundary. Note that the keel does not melt and simulations occur in a reference frame that moves with the keel, that is, $\partial_t \phi = 0$. The geometries of ice keels in the Arctic Ocean are rather complicated and can vary greatly due to the nature of their formation. Computational models typically employ cosine, Gaussian, triangular, and Versoria functions to define the shapes of keels (Zhang et al., 2022). To maintain consistency with previous ice-keel simulations (Skyllingstad et al., 2003; Mortikov, 2016; Pite et al., 1995), we use a Versoria shape of the form

$$
H(x) = \frac{h\sigma^2}{\sigma^2 + 4(x-\ell)^2}, \tag{6}
$$

where $h$ is the maximum ice keel draft, $\sigma$ is the characteristic width of the keel, and $\ell = 75z_0$ is the center of the keel.

At the left and right ends of the domain, sponge layers (identified by their prefactors $\psi$ in Eqs. (1)–(3)) restore the horizontal velocity of the fluid $u$ to a constant $U$, the salinity $S$ to $S_i$, and damp the vertical velocity $w$. We set $\psi = 1$ for $x < 2.5z_0$ and for $x > 117.5z_0$, and $\psi = 0$ in between, with sigmoidal transitions over distances $\varepsilon$, similar to $\phi$. The sponge layers are linearly accelerated from rest to a horizontal speed $U$ in the first $15$ min of simulation time, and then maintained at $U$ for the rest of the simulation ($\approx 35 - 85$ min, see below for details). At approximately $30$ min in each simulation, the upstream sponge layers emit a small disturbance as a result of our implementation of the acceleration, akin to adding noise to simulations or the real ocean. Due to the nature of the two-layered flow, the disturbance amounts to a perturbation in the fluid interface, which travels downstream and, in some cases, triggers fluid instabilities along the way. Due to this ability to "ignite" unstable flows, we refer to this disturbance as the "pilot light", which we will use in § 3.1.

We solve Eqs. (1)–(4) with two-dimensional numerical simulations created in Dedalus, a spectral partial differential equation solver (Burns et al., 2020). We use a Fourier basis in the horizontal, making the domain horizontally periodic. In the vertical, the quantities $u$, $p$, $S$ and $\phi$ are decomposed on a cosine basis, producing homogeneous Neumann boundary conditions for these quantities (*e.g.*, $\partial_z u|_{z=0} = 0$), while $w$ is decomposed on a sine basis, resulting in homogeneous Dirichlet boundary conditions ($w|_{z=0} = 0$). Note that, because the velocity is zero inside the keel, our setup imposes an effective no-slip boundary condition along the ice-water interface. The terms on the left-hand side (right-hand side) of Eqs. (1)–(3) are treated implicitly (explicitly) with a 3rd-order semi-implicit BDF time-stepping scheme (Wang and Ruuth, 2008).

## 2.2 Parameter Space and Experimental Strategy

From a fluid dynamics point of view, density mixing by ice keels falls into the rather well-studied area of two-layer flow over bottom topography (e.g. Baines, 1984; Holland et al., 2002). Similarly to our setup, these previous studies consider an





obstacle that moves through two-layer fluid, causing stirring and, when miscible, irreversible mixing of the two initial densities.
These previous studies indicated that one of the most important parameters characterizing the flow of this system is the Froude number, which is the ratio between the obstacle speed relative to the upper layer and the interfacial wave speed.

When the Froude number is low (typically less than unity) — that is, when the keel is relatively slow or the density stratification is relatively strong — waves can propagate upstream and the resultant flow is called subcritical. Conversely, large Froude numbers (typically larger than unity) correspond to supercritical flows, and any flow feature such as a wave or vortex can only
be swept downstream.

Based on a "bulk" Froude number, Baines (1984) further defined regimes based on the presence of flow characteristics. This included determining whether the flow was supercritical or subcritical, whether transitions between the two occurred, and whether disturbances such as bores or waves were able to propagate upstream. We will find some of these characteristics and regimes in § 3.

Here, we examine five variables that govern the dynamics of the system: keel draft $h$, keel width $\sigma$, mixed layer depth $z_0$, buoyancy difference between the mixed layer and the deeper ocean $\Delta B = g(\rho_2 - \rho_1)/\rho_1$, and horizontal keel speed relative to water $U$. Three independent non-dimensional parameters are sufficient to completely describe the dynamics controlled by these variables. Similarly to previous studies (e.g. Baines, 1984; Cummins, 1995; Houghton and Kasahara, 1968), we choose

$$Fr = \frac{U}{\sqrt{z_0 \Delta B}}, \quad \eta = \frac{h}{z_0}, \quad \text{and} \quad \alpha = \frac{\sigma}{h}, \tag{7}$$

namely, the bulk Froude number, the non-dimensional keel draft, and the keel's aspect ratio, respectively. In our experiments, we will only vary two of these non-dimensional parameters, namely $Fr$ and $\eta$. For each experiment, we set the aspect ratio $\alpha = 3.9$, based on observations showing that this value is consistent among first-year ice floes (no such consistency is apparent in multi-year ice) (Timco and Burden, 1997).

To obtain the permissible ranges of values for $Fr$ and $\eta$, we establish bounds from observations of the Arctic Ocean. Ice keel
drafts are, on average, $7.45\,\mathrm{m}$, but can reach depths of up to $45\,\mathrm{m}$ (Wadhams, 2012; Kvadsheim, 2014). For the depth of the mixed layer, we take $8\,\mathrm{m} \leq z_0 \leq 22.4\,\mathrm{m}$, which is consistent with the observed range of summer mixed-layer depths across the Arctic Ocean (Peralta-Ferriz and Woodgate, 2015). In the marginal ice zone, Cole et al. (2017) estimate $U \leq 0.4\,\mathrm{m\,s^{-1}}$. Lastly, we calculate the bounds on $\Delta B$ using the bounds on $\Delta S$ between summer and winter mixed layers reported by Peralta-Ferriz and Woodgate (2015) and use EOS-80 to calculate the respective densities. With the upper and lower bounds on $\rho_1$ and $\rho_2$ thus
calculated, we obtain $3.1 \times 10^{-3}\,\mathrm{m\,s^{-2}} \leq \Delta B \leq 7.5 \times 10^{-2}\,\mathrm{m\,s^{-2}}$. Thus, we find that

$$0 \leq Fr \leq 2.5 \qquad \text{and} \qquad 0 \leq \eta \leq 5.6. \tag{8}$$

Simulations with $Fr = 2.5$ could not run throughout the entire $\eta$ range, nor could simulations with $\eta > 2$ run throughout the entire $Fr$ range, without encountering numerical instabilities. We, therefore, consider four values of $\eta$, namely $0.50$, $0.95$, $1.2$ and $2.0$, and four values of $Fr$, namely $0.5$, $1.0$, $1.5$ and $2.0$. We name our simulations based on their $Fr$ and $\eta$ values,
following the convention $\mathrm{F}\lfloor 10Fr \rfloor \mathrm{H} \lfloor 10\eta \rfloor$, where $\lfloor \cdot \rfloor$ is the floor operator. For example, we refer to the $Fr = 0.5$, $\eta = 0.95$ simulation as F05H09.



To vary $Fr$ and $\eta$, we set $z_0 = 8$ m, $\Delta B = 0.015$ m s$^{-2}$, $S_1 = 28$ psu, and $S_2 = 30$ psu and, therefore, vary $U$ and $h$ through the respective ranges $0.17$ m s$^{-1} \leq U \leq 0.69$ m s$^{-1}$ and $4$ m $\leq h \leq 16$ m, according to Eq. (7). These parameter choices yield Reynolds numbers $Re = Uh/\nu$ ranging from $O(10^2)$ to $O(10^3)$. Note that we increase the phase field steepness parameter $\varepsilon$
for $\eta = 2.0$ simulations to prevent salinity leakage from the keel. See Table 1 for a summary of the parameter values for all simulations.

The simulation domain has height $10z_0$ and length $120z_0$. We focus our analysis on regions upstream and downstream of the keel, which we choose to be $20z_0 \leq x \leq \ell$ and $\ell \leq x \leq 115z_0$, respectively, and that occupy the entire water column (see Figure 1). For simulations with $Fr < 2.0$, a stagnant layer forms upstream of the keel and its spatial extent grows over time
in the wake of an interfacial wave. This wave eventually reflects against the edge of the upstream sponge layer and propagates downstream. We consider processes that occur after this wave reaches the upstream region at $x = 20z_0$ to be an artifact of our setup, marking the time when we stop processing data. We found that this time window varied more directly with $Fr$ than with $\eta$. Consequently, measuring simulation time in units of $t_0 = \sqrt{z_0/\Delta B}$ ($\approx 23$ s), we stopped simulations with $Fr = 1.5$ and $Fr = 2.0$ at $t = 270t_0$, simulations with $Fr = 1.0$ at $t = 156t_0$, and simulations with $Fr = 0.5$ at $t = 132t_0$. Each simulation
uses a time step of $dt = 6 \times 10^{-3}$ s $\approx 2.6 \times 10^{-4}t_0$ and a domain discretized with $1280$ horizontal and $640$ vertical grid points.

In some experiments, we observe vertical aliasing in the mixed and deep layers in the form of horizontal alternating bands in the spanwise vorticity $q$ and vertical density gradients $\partial_z \rho$. This aliasing has no effect on the stability of the numerical integration. It does create artificially enhanced density mixing values in weakly stratified regions that we can easily remove from our diagnostic calculations (see below).

## 2.3 Quantifying Mixing

To analyze the vertical mixing induced by the motion of ice keels, it is necessary to distinguish mixing from stirring. Mixing is the irreversible smoothing out of density gradients that may exist across isopycnals by molecular diffusion of density. Stirring, on the other hand, is the rearrangement of fluid parcels induced by fluid motion, which can sharpen diapycnal density gradients and increase the surface area that density can diffuse across. Stirring can therefore lead to mixing but is, in principle, reversible.

Both mixing and stirring alter the average total potential energy per unit mass of a system, which we define at any given time $t$ as

$$\mathscr{P} = \frac{1}{\rho_1 A_\Omega} \iint_\Omega g\rho(x, z, t)z \, dx \, dz, \tag{9}$$

where $\Omega$ is the integration domain, representing the upstream or downstream region sketched in Fig. 1, and $A_\Omega = \iint_\Omega dx \, dz$ is its surface area.

To separate irreversible mixing from reversible stirring, we partition $\mathscr{P}$ into available ($\mathscr{P}_a$) and background ($\mathscr{P}_b$) average potential energies per unit mass such that $\mathscr{P} = \mathscr{P}_a + \mathscr{P}_b$ (Lorenz, 1955). $\mathscr{P}_b$ is defined as the lowest potential energy state available to the system, which is given at any time $t$ as

$$\mathscr{P}_b = \frac{1}{\rho_1 A_\Omega} \iint_\Omega g\rho_*(z, t)z \, dx \, dz, \tag{10}$$





**Table 1.** The values of all parameters for all sixteen simulations. $\varepsilon$ is the phase field steepness parameter, $Re$ is the Reynolds number, and $t_{\mathrm{rt}}$ is the total run time of the simulation.

| $Fr$ | $\eta$ | Simulation name | $\varepsilon$ [m] | $t_{\mathrm{rt}}$ [$t_0$] | $Re$ |
|------|--------|------------------|-------------------|---------------------------|------|
| 0.5 | 0.5 | F05H05 | 0.125 | 132 | 340 |
| 0.5 | 0.95 | F05H09 | 0.125 | 132 | 646 |
| 0.5 | 1.2 | F05H12 | 0.125 | 132 | 816 |
| 0.5 | 2.0 | F05H20 | 0.140 | 132 | 1360 |
| 1.0 | 0.5 | F10H05 | 0.125 | 156 | 680 |
| 1.0 | 0.95 | F10H09 | 0.125 | 156 | 1292 |
| 1.0 | 1.2 | F10H12 | 0.125 | 156 | 1632 |
| 1.0 | 2.0 | F10H20 | 0.140 | 156 | 2720 |
| 1.5 | 0.5 | F15H05 | 0.125 | 270 | 1040 |
| 1.5 | 0.95 | F15H09 | 0.125 | 270 | 1976 |
| 1.5 | 1.2 | F15H12 | 0.125 | 270 | 2496 |
| 1.5 | 2.0 | F15H20 | 0.135 | 270 | 4160 |
| 2.0 | 0.5 | F20H05 | 0.125 | 270 | 1380 |
| 2.0 | 0.95 | F20H09 | 0.125 | 270 | 2622 |
| 2.0 | 1.2 | F20H12 | 0.125 | 270 | 3312 |
| 2.0 | 2.0 | F20H20 | 0.135 | 270 | 5520 |

where $\rho_*$ is the sorted density field at time $t$. Note that $\rho_*$ at each time step is only a function of depth $z$, and can be thought of as the reference density at a given depth if the water parcels in $\Omega$ were sorted by density (with the densest parcels at the bottom) to achieve a state of minimum potential energy. An inverse quantity is $z_*(\rho(x,z,t))$, which is the reference depth at which a water parcel with density $\rho$ would reside after sorting. At each time $t$, $z_*$ is uniquely a function of density $\rho$. Reference depth and density, respectively $z_*$ and $\rho_*$, are related by $z_*(\rho_*(z,t)) = z$.

Fick's law for salt diffusion implies that molecular diapycnal density fluxes make denser waters irreversibly lighter, and vice versa. These molecular processes result in denser waters that end up at depths higher than their original depth because they become less salty and vice versa, thus changing the sorted density profile $\rho_*$ in time. This scenario irreversibly raises the center of gravity of the sorted density profile, increasing $\mathscr{P}_b$. We can measure this rise following the convention of Winters et al. (1995) and Salehipour and Peltier (2015) by defining

$$\frac{\mathrm{d}\mathscr{P}_b}{\mathrm{d}t} = S_{\mathrm{diff}} + S_{\mathrm{adv}} + \Phi_\Omega, \tag{11}$$



where $S_{\text{diff}}$ and $S_{\text{adv}}$ are the rates of change of $\mathscr{P}_b$ due to diffusive and advective transfers of potential energy across the boundary of $\Omega$, respectively, and

$$\Phi_\Omega = \frac{\mu g}{\rho_1 A_\Omega} \iint\limits_\Omega |\nabla\rho|^2 \frac{\mathrm{d}z_*}{\mathrm{d}\rho}\, \mathrm{d}x\,\mathrm{d}z \tag{12}$$

is the average irreversible mixing rate within $\Omega$, that is, the average background potential energy flux due to irreversible diapycnal processes alone.

Following Salehipour and Peltier (2015), we can treat $\Phi_\Omega$ as an average buoyancy flux within $\Omega$ that is the product of the vertical buoyancy gradient associated with the sorted density profile and the diffusivity, which is the coefficient that measures the medium's ability to "smooth out" said gradient. Because diffusivity is arguably more intuitive and is often parameterized in climate models (*e.g.*, Large et al., 1994), we also compute the spatially averaged diapycnal diffusivity, namely,

$$K_\Omega = \frac{1}{\mu}\frac{\Phi_\Omega}{N_{*,\Omega}^2} \qquad \text{with} \quad N_{*,\Omega}^2 = -\frac{g}{\rho_1 A_\Omega}\iint\limits_\Omega \frac{\mathrm{d}\rho_*}{\mathrm{d}z}\,\mathrm{d}x\,\mathrm{d}z, \tag{13}$$

Note that because of the division by $\mu$, $K_\Omega$ is non-dimensional.

Finally, we introduce $Z_\Omega$, the typical depth (in units of $z_0$) above which 95% of the time-averaged mixing occurs. Specifically, we define $Z_\Omega$ such that

$$\int\limits_0^{Z_\Omega}\int\limits_{L_\Omega(z)} |\nabla\rho|^2 \frac{\mathrm{d}z_*}{\mathrm{d}\rho}\,\mathrm{d}x\,\mathrm{d}z = 0.95\Phi_\Omega, \tag{14}$$

where $L_\Omega(z)$ is the $x$-interval within $\Omega$ at depth $z$.

We compute $\Phi_\Omega$ in the upstream ($\Phi_U$) and downstream ($\Phi_D$) regions separately and temporally average them from $t = 81t_0$ to $t = t_{\text{rt}}$ (recall Table 1), denoting the resulting quantities with an overline. The acceleration period ends at $t = 42t_0$, allowing sufficient time for any transient behavior to disappear before averaging. This yields the quantities $\overline{\Phi}_U$ and $\overline{\Phi}_D$, and we define $\overline{K}_U$, $\overline{K}_D$, $\overline{Z}_U$, and $\overline{Z}_D$ following the same procedure.

Small amounts of artificially large salinity values ($\rho > \rho_2$), perhaps due to aliasing, are created near the keel and travel deeper into the domain. To ensure that this does not interfere with our computation of $\Phi_\Omega$, we need to remove the resulting artificial density gradients. We set values of $|\nabla\rho|^2$ less than $3\times10^{-6}(\Delta\rho/b)^2$ to 0 (recall $b$ from Eq. 5). This does not alter our estimates of $\Phi_\Omega$ near the pycnocline, where most of the physical mixing occurs with $|\nabla\rho|^2 = O(10^{-3} - 10^{-5})(\Delta\rho/b)^2$.

The concepts described in this section apply only when the equation of state is linear, that is, when $\rho(S) \propto S$ (Winters et al., 1995; Tailleux, 2009). We verified that, in our simulations, this linearity condition is satisfied to a very good approximation.

## 3 Stirring Regimes

In order to understand the mechanisms driving vertical mixing by ice keels, we begin by classifying the upstream and downstream flow behavior of each simulation into different stirring regimes. We qualitatively determine each category based on





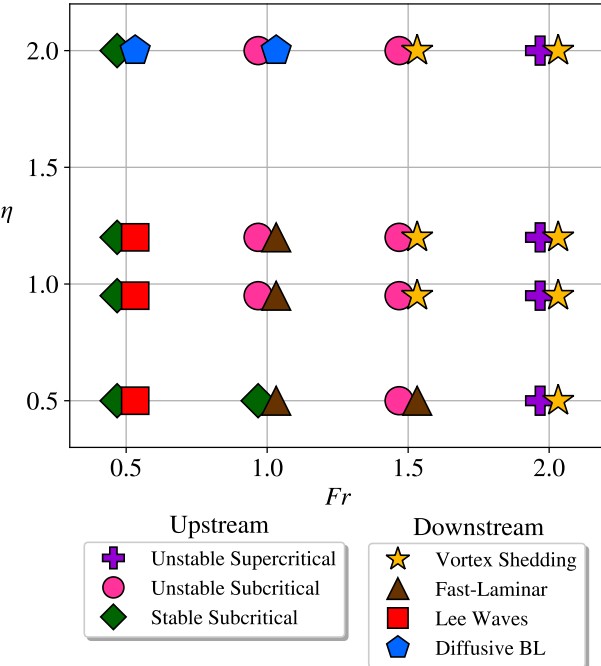

**Figure 2.** Upstream and downstream regime classification for each simulation in dimensionless parameter space $(Fr, \eta)$. The upstream and downstream markers are offset to the left and right, respectively, on the $Fr$ axis for legibility.

recurring density structures and characteristic flow patterns in the $(Fr, \eta)$ space. Our categorization merely aims to provide a narrative framework for the mixing measurements of § 4. Consequently, the number of categories and the delineations between

240 each are somewhat subjective. We summarize our classification in Figure 2, and provide more detailed descriptions in the remainder of this section. Note that we provide videos for every simulation in the supplementary material.

## 3.1 Upstream Regimes

In the upstream region, we identify three flow regimes: "Unstable Supercritical", "Unstable Subcritical", and "Stable Subcritical". Figure 3 shows one representative snapshot of a set of streamlines, superposed on the density field upstream, for each

245 regime.

Although the Sub- and Supercritical qualifiers usually relate to the local Froude number (recall § 2.2), we do not measure it. Instead, we visually identify simulations where we observe upstream propagation of disturbances away from the keel and call them "Subcritical", and call the others "Supercritical". This may lead to marginal disagreements with a stricter local Froude-based categorization. Indeed, the upstream disturbances we observe are sometimes non-linear, solitary-looking waves

250 that might propagate slightly faster than the linear propagation speed (Helfrich and Melville, 2006), the latter being part of the Froude number's definition. It is therefore possible, in principle, to see non-linear disturbances propagate upstream in a





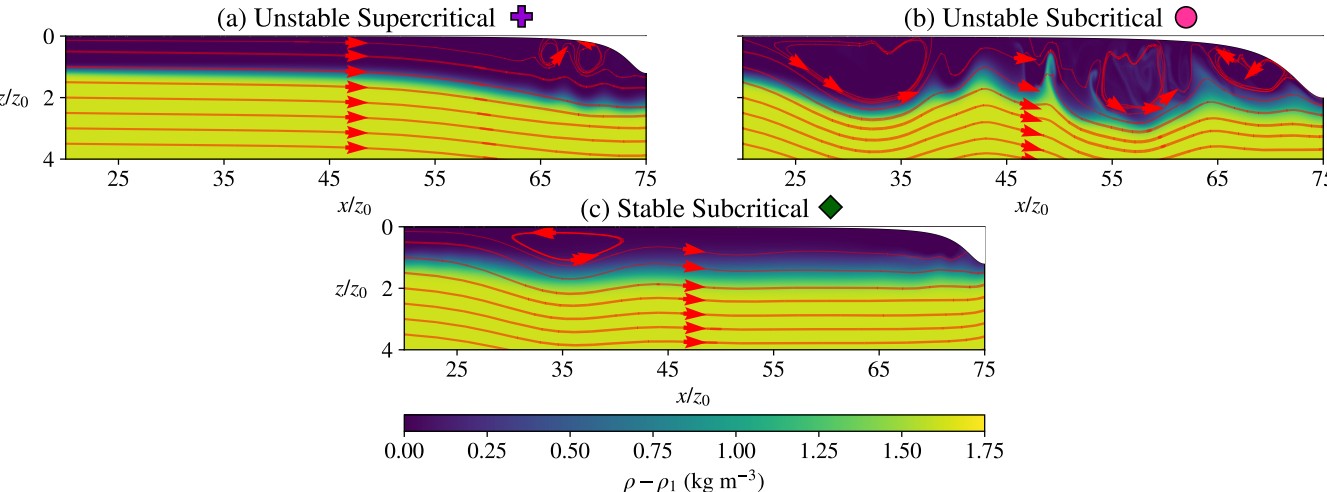

**Figure 3.** Representative snapshots of the three upstream flow regimes' density field $\rho$ overlaid with velocity streamlines. Each regime is assigned a marker in the title of the panel that will be used in the figures for the remainder of the paper. (a) "Unstable Supercritical" snapshot with F20H12 at $t = 246t_0$; (b) "Unstable Subcritical" with F10H20 at $t = 107t_0$; (c) "Stable Subcritical" with F05H12 at $t = 90t_0$.

supercritical flow. We choose to ignore these subtle theoretical considerations and base our classification on whether we can visually identify upstream wave propagation.

The qualifiers "Stable" and "Unstable" indicate whether small perturbations upstream of the keel eventually develop into a vigorous vorticity field, which can then cause significant stirring. We will not try to characterize or identify the instability at play, nor do we even claim that there is a single instability. When instabilities arise, they often grow from the boundary layer in contact with the keel, and their onset often coincides with the passage of the "pilot light" mentioned in § 2.2. We find the "pilot light" useful in probing the stability of the flow to perturbations. Note that in some low-$Fr$ cases, we stop the simulation before the "pilot light" has had time to reach the keel. In those cases however, it dissipates very early without triggering any instability. Regardless of their origin, disturbances often amplify when they coincide with large-scale pycnocline oscillations, as seen in the relevant video supplements. For the purposes of this work, we simply notice when vorticity grows enough to vigorously stir the flow, and name our regimes accordingly.

The **Unstable Supercritical regime** (Figures 3a and 4) only occurs in simulations with $Fr = 2.0$, our largest value for this parameter. Away from the keel, we do not observe noticeable stirring of the interface. Upstream of the keel, the viscous boundary layer generates vorticity that eventually travels under the keel and downstream, as the vortices in Figure 4b are about to do. We note that a vigorous vorticity field does not always equate to mixing-inducing stirring, since the vortices in this figure mostly stir mixed-layer fluid and leave the pycnocline smooth.

The **Unstable Subcritical regime** (Figures 3b and 5) occurs for $Fr = 1.5$ and all but one of the simulations with $Fr = 1.0$. Recall that $Fr$ is merely a bulk estimate of the local Froude numbers and does not indicate supercritical flow. In this regime, we see large-scale pycnocline oscillations that sometimes resemble solitary waves and sometimes are more turbulent, Figure 3b





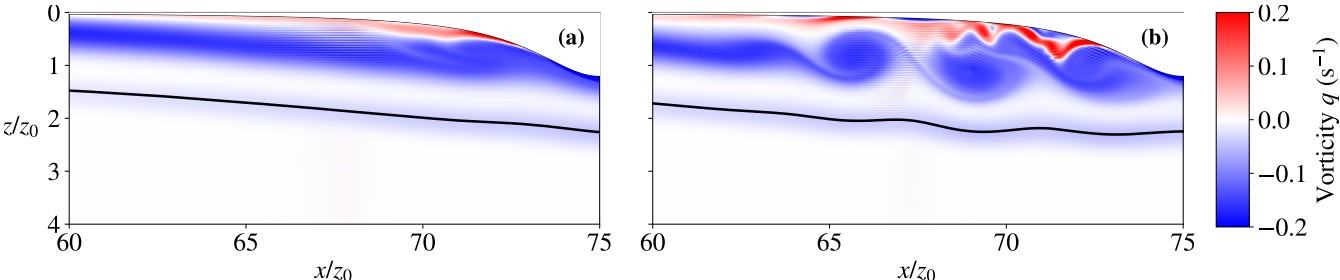

**Figure 4.** Spanwise upstream vorticity $q$ at (a) $t = 187t_0$ and (b) $t = 246t_0$ for the F20H12 simulation in the Unstable Supercritical regime. The solid black line is the $(\rho_1 + \rho_2)/2$ contour, which marks the center of the pycnocline.

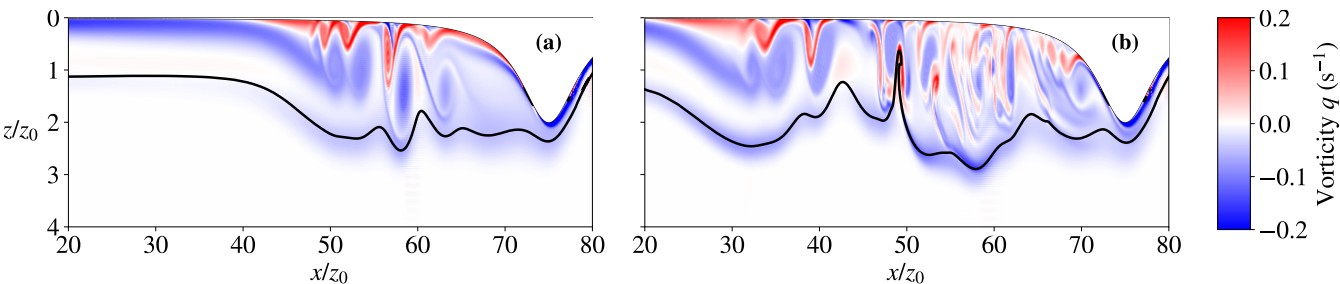

**Figure 5.** Spanwise upstream vorticity $q$ at (a) $t = 71t_0$ and (b) $t = 107t_0$ for the F10H20 simulation in the Unstable Subcritical Waves regime. The solid black line is the $(\rho_1 + \rho_2)/2$ contour, which marks the center of the pycnocline.

showing a hybrid example. These disturbances correlate with significant vorticity generation ahead of the keel, as we can see in Figure 3b for $40 < x/z_0 < 70$. In the wake of these pycnocline oscillations, significant small-scale flows stir the fluid and sometimes entrain water from the deeper ocean into the mixed layer, which induces mixing (Fig. 5b at $x/z_0 \approx 50$). The mean flow then sweeps these vortices downstream towards the keel, some lasting long enough to pass under it.

The **Stable Subcritical regime** (Figure 3c) occurs in all simulations with $Fr = 0.5$ and in F10H05. In this regime, we observe upstream-propagating large-scale perturbations that resemble solitary waves in F05H09 and F05H12, and appear more linear in nature in F05H05 and F10H05. In any case, the flow remains stable to perturbations, and this regime does not see much stirring.

### 3.2    Downstream Regimes

In the downstream portion of the domain, we identify four flow regimes: "Vortex Shedding", "Diffusive Boundary Layer" "Fast-Laminar", and "Lee Waves". Figure 6 shows one representative snapshot per regime.

The **Vortex Shedding regime** (Figure 6a) is only associated with "Unstable" upstream stirring regimes. Specifically, this regime occurs in all $Fr = 2.0$ simulations, which we identified as Unstable Supercritical in the upstream region, and in simulations with $Fr = 1.5$ and $\eta \geq 0.95$, which we identified as Unstable Subcritical simulations with the largest Froude numbers.





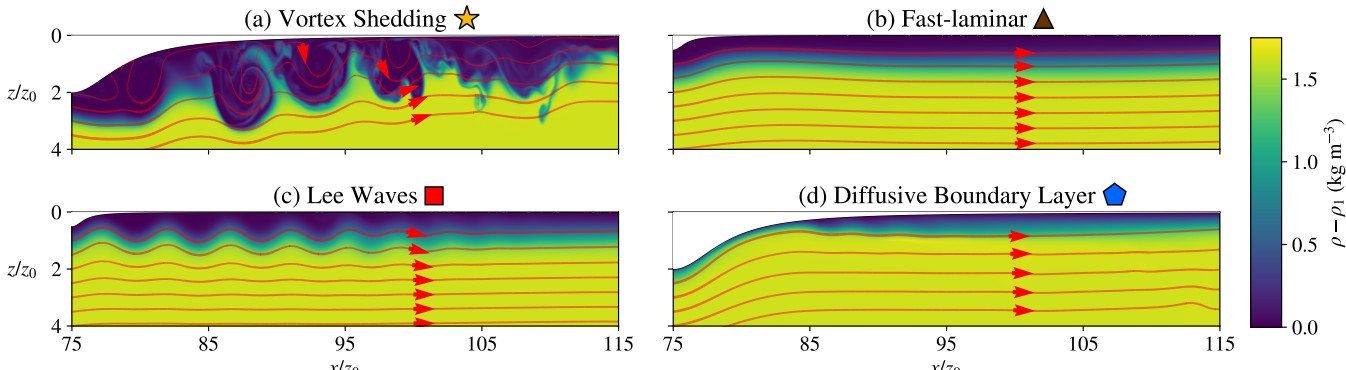

**Figure 6.** Representative snapshots of the four downstream flow regimes' density field $\rho$ overlaid with velocity streamlines. Each regime is assigned a marker in the title of the panel, which will be referenced in the remainder of the paper. (a) Vortex Shedding corresponds to F20H20 at $t = 192t_0$; (b) Fast-laminar to F10H05 at $t = 144t_0$; (c) Lee Waves to F05H05 at $t = 108t_0$; and (d) Diffusive Boundary Layer to F05H20 at $t = 96t_0$.

Here, vortices downstream of the keel, the largest of which have length scales similar to the draft of the keel $h$, stir deeper ocean water into the mixed layer while maintaining large local density gradients. Some vortices grow on the lee side of the keel and some originate from the upstream region (see § 3.1, Figures 4b and 5b). This regime shows phenomena that are the most closely related to turbulent hydraulic jumps.

     We note that Baines (1984) did not observe what we call vortex shedding for $Fr \sim 2$. Instead, their flow remained super-
critical downstream, without noticeable vortices. However, Hester et al. (2021) did observe vortex shedding in their numerical simulations along the bottom crest of their rectangular obstacle due to flow separation at the windward edge of the obstacle (see their Figure 7). As mentioned in § 2.1, our code is based on that of Hester et al. (2021), so it is not surprising that we see similar flow behavior.

     The **Fast-Laminar regime** (e.g., Figure 6b) occurs in F15H05 and for all values of $Fr = 1.0$ except F10H20. Most of
the stirring in this regime occurs when episodic vortices generated upstream are advected under the keel (see, *e.g.*, video for F15H05 at $t \approx 190t_0$).

     The **Lee Waves regime** occurs for small to moderate keel drafts, which do not completely block the mixed layer, and for the smallest Froude number ($Fr = 0.5$). For example, in Figure 6c, we observe a train of lee waves that extends $\sim 30z_0$ in the horizontal direction. Because the flow is slow, the creation of boundary layer vorticity is weak, and perturbations of upstream
disturbances, if they pass the keel at all, do not get amplified. Therefore, stirring remains very weak.

     Lastly, the defining feature of the **Diffusive Boundary Layer** regime is a stream of mixed-layer water creeping along the lee side of the keel, visible for $75 \leq x/z_0 \leq 85$ in Fig. 6d. We observe this regime when the keel draft is maximum and the Froude number is low, namely, in F05H20 and F10H20. Here, we depart slightly from the purely kinematic criteria that we have applied so far to define all other regimes. Indeed, while this regime may not look visually very different from the weak



**Table 2.** Upstream and downstream normalized irreversible mixing rates, diapycnal diffusivities, and depths values for each simulation. $\overline{\Phi}_0 \approx 4.0 \times 10^{-7} \mathrm{W\,kg^{-1}}$ is our reference value (indicated as "REF" in the table).

| Simulation | Upstream | | | Downstream | | |
|---|---|---|---|---|---|---|
| name | $\overline{K}_U$ | $\dfrac{\overline{\Phi}_U}{\overline{\Phi}_0} - 1$ | $\overline{Z}_U$ | $\overline{K}_D$ | $\dfrac{\overline{\Phi}_D}{\overline{\Phi}_0} - 1$ | $\overline{Z}_D$ |
| F05H05 | 0.97 | REF | 1.7 | 1.0 | +0.3% | 1.5 |
| F05H09 | 0.77 | -4.2% | 1.9 | 1.1 | +15% | 1.3 |
| F05H12 | 0.77 | -5.1% | 2.0 | 1.2 | +20% | 1.2 |
| F05H20 | 0.69 | -11% | 2.3 | 2.7 | +170% | 1.7 |
| F10H05 | 0.98 | -1.3% | 1.7 | 0.94 | -6.1% | 1.5 |
| F10H09 | 0.99 | -0.1% | 2.0 | 0.94 | -2.1% | 1.5 |
| F10H12 | 1.0 | +7.4% | 2.2 | 1.0 | +2.7% | 1.6 |
| F10H20 | 1.1 | +50% | 2.9 | 1.7 | +73% | 1.8 |
| F15H05 | 0.92 | -0.3% | 1.8 | 0.96 | -4.7% | 1.7 |
| F15H09 | 1.0 | +3.9% | 2.3 | 1.1 | +7.8% | 2.1 |
| F15H12 | 1.1 | +12% | 2.6 | 1.4 | +37% | 2.2 |
| F15H20 | 0.94 | +47% | 3.2 | 1.5 | +84% | 2.9 |
| F20H05 | 0.91 | +0.8% | 1.6 | 1.0 | +0.4% | 1.9 |
| F20H09 | 0.98 | +0.4% | 1.9 | 2.0 | +109% | 2.2 |
| F20H12 | 0.94 | -2.0% | 2.1 | 3.1 | +225% | 2.4 |
| F20H20 | 0.76 | +39% | 2.6 | 7.0 | +805% | 3.0 |

stirring seen in the Fast-Laminar and Lee-Wave regimes, we will see in §4 that the mixing induced by the flow warrants its own category.

## 4 Diapycnal Mixing

We now examine the distribution of diapycnal mixing ($\overline{\Phi}_U$, $\overline{\Phi}_D$, $\overline{K}_U$, and $\overline{K}_D$) and their typical mixing depths ($\overline{Z}_U$ and $\overline{Z}_D$) across all simulations, as summarized in Table 2. Below, we use our stirring regime categorization to (1) interpret variations across the simulations and (2) interpret differences between simulations that are not consistent with the null hypothesis that simulations with faster (larger $Fr$) and deeper (larger $\eta$) keels facilitate more diapycnal mixing (larger $K$, $\Phi$) that extend over a larger depth range (larger $Z$).

Figure 7 shows the irreversible mixing rates $\overline{\Phi}_U$ and $\overline{\Phi}_D$ for each simulation. Note that $\overline{K}_U$ ($\overline{K}_D$) follows trends similar to those of $\overline{\Phi}_U$ ($\overline{\Phi}_D$), as can be pieced together from Table 2. To keep the presentation simple, we therefore focus our figures on $\overline{\Phi}_U$ and $\overline{\Phi}_D$. We compare all of these values with $\overline{\Phi}_0 \approx 4.0 \times 10^{-7} \mathrm{W\,kg^{-1}}$, the value of $\overline{\Phi}_U$ in F05H05, whose flow is least





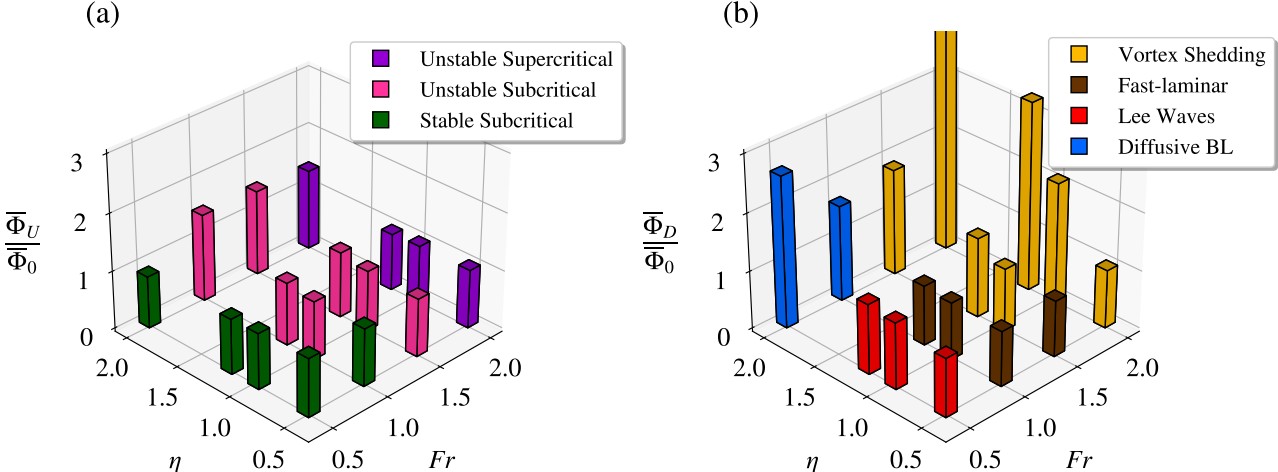

**Figure 7.** Time-averaged non-dimensional diapycnal mixing rate (a) upstream $\overline{\Phi}_U$ and (b) downstream $\overline{\Phi}_D$ for each simulation. For F20H20, $\overline{\Phi}_D \approx 9.0\overline{\Phi}_0$.

disturbed by the keel of all experiments. This does not imply that $\overline{\Phi}_0$ is the smallest value of all the mixing rates we measure, as we are about to see.

Overall, **upstream of the keel**, we find fairly similar mixing rates $\overline{\Phi}_U$ across the simulations; 11 of the 16 simulations have values of $\overline{\Phi}_U$ that differ from $\overline{\Phi}_0$ by less than 10%. We obtain comparably small mixing values possibly because the density

gradient at the pycnocline is too large to allow the growth of instabilities within the pycnocline that would be required to mix it, and such instabilities are indeed not observed upstream of the keel in most simulations. This point will be discussed in further detail in § 5. The remaining five simulations have the deepest keels ($\eta = 2.0$), except F15H12, and differ by up to 50% relative to $\overline{\Phi}_0$. We note that the simulation with the largest mixing rate does not occur in the deepest and fastest keel.

For a given $Fr$, simulations tend to belong to the same stirring regime that we identified in § 3. Mixing rates $\overline{\Phi}_U$ increase

with $\eta$ as per our null hypothesis for $Fr = 1.0, 1.5$ (Unstable Subcritical regime), but not for $Fr = 0.5$ (Stable Subcritical regime) and $Fr = 2.0$ (Unstable Supercritical regime). At a given $\eta$ across $Fr$, we find that trends in $\overline{\Phi}_U$ largely do not satisfy our null hypothesis, except for the Unstable Subcritical regime. These findings are described in greater detail below.

First, we find two instances where simulations with faster keels have less mixing than simulations slower keels. Specifically, $Fr = 2.0$ simulations with deeper keels (F20H12 and F20H20) have $\overline{\Phi}_U$ values 6-12% smaller than their $Fr = 1.5$ and $Fr = $

1.0 counterparts (F10H12, F10H20, F15H12 and F15H20). This occurs because vortices tend to remain trapped against the keel in the thicker mixed layer that develops in simulations with deeper keels. Fig. 4 shows a typical configuration for one Unstable Supercritical simulation ($Fr = 2.0$), where the vortices remain close to the keel, primarily stirring the already-mixed fluid, and get swept under the keel before causing much mixing in the upstream region. Sometimes deeper vortices that extend into the deeper ocean layer do develop and cause turbulent mixing, but occur rather infrequently leading to an insignificant

contribution to the average mixing rates.



In contrast, the deeper keel simulations with smaller $Fr = 1.0, 1.5$ have larger vortices that occur more frequently, are able to sometimes pull deep water into the upper layer (see, *e.g.*, Figs. 3b and 5b around $x/z_0 \approx 50$ taken from F10H20) and remain in the upstream region longer (consistent with the subcritical flow categorization), which induces more mixing upstream. In particular, Fig. 3b shows an example of the stronger density gradients that are induced by perturbations in the interface in these

simulations in the Unstable Subcritical regime. These density perturbations get larger with both $\eta$ and $Fr$, resulting in trends in $\overline{\Phi}_U$ that are consistent with our null hypothesis.

Second, considering only simulations with $Fr = 0.5$ (Stable Subcritical Regime), we find that simulations with deeper keels have less mixing than those with shallower keels. For example, the simulation with the shallowest keel (F05H05) has a $\overline{\Phi}_U$ value 11% larger than the simulation with the deepest keel (F05H20). Although we did not investigate this result in detail,

visual inspection of the videos suggests that this is caused by the tendency for deep keels to block more flow. The blocked flow causes the distance between two isopycnals in the upstream stagnant layer to increase, implying smaller density gradients and less mixing.

Overall, **downstream of the keel**, we find more variation between simulations than upstream of the keel, with only 7 out of 16 simulations yielding mixing rates $\overline{\Phi}_D$ that differ by less than 10% from $\overline{\Phi}_0$. Of the remaining nine simulations, six have

$\overline{\Phi}_D$ that are at least 73% larger than $\overline{\Phi}_0$, and three simulations have $\overline{\Phi}_D > 2\overline{\Phi}_0$. We note that these three simulations are in the downstream Vortex Shedding and upstream Unstable Supercritical regimes, implying that all vortices formed upstream are pushed downstream. Notably, the vortex shedding events are episodic. As such, for F20H05, in which only one vortex shedding event occurs over our averaging time interval, $\overline{\Phi}_D \sim \overline{\Phi}_0$. During the event, mixing is enhanced with rates up to $1.5\overline{\Phi}_0$.

For a given $Fr$, we find that deeper keels do indeed mix more and therefore satisfy the null hypothesis. Furthermore, when

$Fr \geq 1.0$, faster keels (increasing $Fr$) mix more for a given keel height $\eta$, also in line with the null hypothesis. On the other hand, there are several instances where simulations with faster keels have less mixing than those with slower keels. Specifically, for a given $\eta$, $\overline{\Phi}_D$ decreases when $Fr$ increases from 0.5 to 1.0. Based on our stirring regime categorization, this appears to occur for either of two reasons: First, for $\eta \leq 1.2$, in the Lee Wave regime of the $Fr = 0.5$ simulations (F05H05, F05H10 and F05H12) the isopycnals are spatially perturbed, resulting in larger density gradients and larger mixing compared with the

Fast-Laminar regime ($Fr = 1.0$), where isopycnals are predominantly flat (cf. Fig. 6b,c). In the Fast-Laminar regime, mixing is episodically enhanced by passing vortices that are generated and advected from the upstream, but these isopycnal perturbations and thus induced mixing are small at $Fr = 1.0$. Second, the F05H20 to F10H20 transition at $\eta = 2.0$, which occurs in the Diffusive BL regime, also sees a drop in $\overline{\Phi}_D$. In this regime, most of the mixing happens in the diffusive BL against the lee side of the keel. This boundary layer is much thinner in F05H20 than in the faster flowing F10H20, whereas the buoyancy difference

$\Delta B$ across both boundary layers is always the same. The net result, after spatial integration, is more mixing in F05H20 than in F10H20. We leave a more quantitative analysis of these deviations to future studies.

In addition to the variability in mixing rates, we are also interested in the **mixing depth**, that is, how deep mixing penetrates relative to the pycnocline depth depending on keel height and speed. Figure 8 displays $\overline{Z}_U$ and $\overline{Z}_D$ for each experiment. Recall from § 2.3 that $Z_\Omega$, with $\Omega = U$ or $D$, is the depth that captures 95% of the integrand of $\Phi_\Omega$ in Eq. (12), and therefore provides

an estimate of how deep we see active mixing. On average, we find that vertical mixing extends approximately twice as deep as





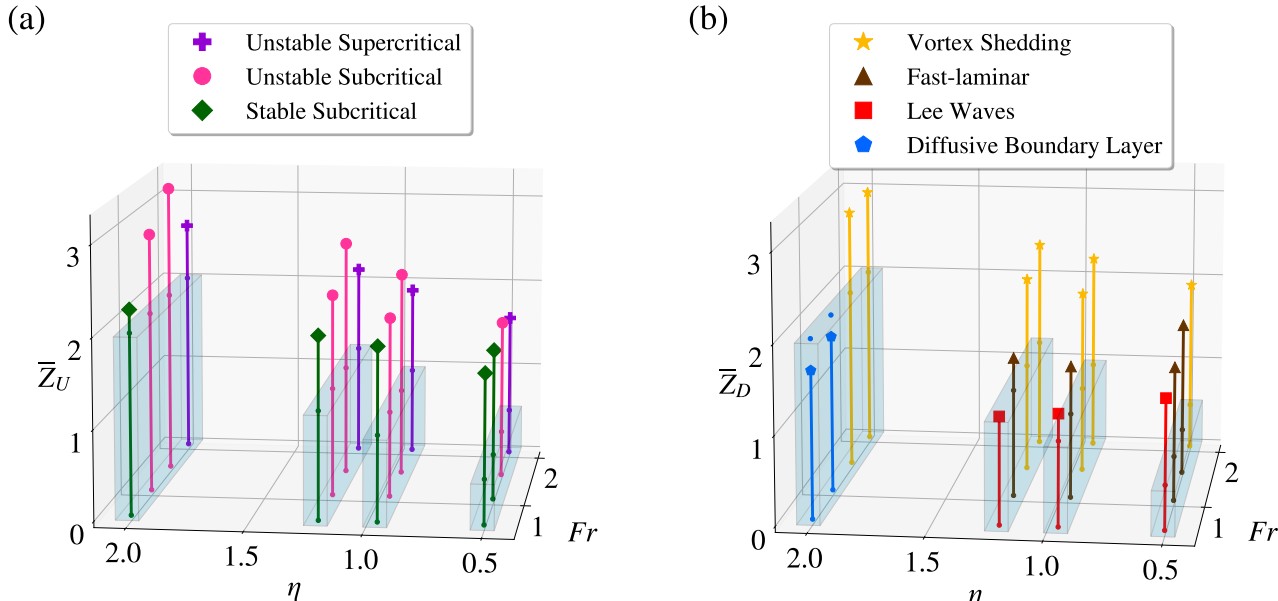

**Figure 8.** (a) $\overline{Z}_U$ and (b) $\overline{Z}_D$ for each simulation. The transparent rectangular boxes mark the non-dimensional keel drafts $\eta$, and the small round markers mark the depths $z = 0$ and $z/z_0 = \eta$.

the initial depth of the mixed layer, both upstream and downstream. Specifically, we find that upstream, $1.6 \leq \overline{Z}_U \leq 3.2$, with an average $\overline{Z}_U = 2.2 \pm 0.5$ and downstream $1.2 \leq \overline{Z}_D \leq 3.0$, with an average $\overline{Z}_D = 1.9 \pm 0.5$ (where $\pm$ indicates one standard deviation).

For most simulations, in the upstream and downstream, we find that the mixing depth increases both with keel speed (larger $Fr$) and keel height (larger $\eta$). However, the comparison of the upstream mixing depth values in the simulations with $Fr = 2.0$ to $Fr = 1.5$ indicates that flows induced by faster keels of the same height can result in mixing that occurs over a shallower depth range. This possibly occurs because $Fr = 1.5$ simulations have a subcritical flow and, therefore, can trap vortices upstream, which can ultimately deepen the range of mixing. In contrast, $Fr = 2.0$ simulations have a supercritical flow and vortices, which could otherwise increase $\overline{Z}_U$, are swept downstream. Indeed, downstream of the keel, where such vortices enhance mixing, mixing depth $\overline{Z}_D$ is larger in simulations with $Fr = 2.0$ than in those with $Fr = 1.5$.

We find that for almost all of the simulations mixing extends below the keel depth both upstream and downstream (i.e., $\overline{Z}_U, \overline{Z}_D > \eta$) with the exception of the downstream in F05H20 and F10H20. These two simulations fall in the Diffusive Boundary Layer downstream regime, in which mixing predominantly occurs along the lee-side boundary of the keel. This result, along with the trend in mixing rate $\overline{\overline{\Phi}}_D$ that is contrary to our null hypothesis for these two simulations (Fig. 7b), supports our classification of these simulations in their own regime even though their streamlines may not look substantially different from the Fast Laminar regime (cf. Fig. 6).





We caution, however, against over-interpreting these numbers. Indeed, differences between them are often smaller than the standard deviation of the mixing depth for a given simulation over the time average. See Discussion for more on this point.

## 5  Discussion

### 5.1  Implications

We have presented results examining how the size and speed of an ice keel impacts ocean mixing using idealized simulations with different ice keel characteristics (different values of $\eta$ and $Fr$). To put our findings into context, we use observations reported by previous studies to roughly estimate climatological averages and decadal trends of $\eta$ and $Fr$ in the Arctic Ocean. Specifically, we select climatological averages and decadal trends of $U$, $h$, $\Delta B$, and $z_0$ to estimate climatological values and decadal trends of $\eta$ and $Fr$. The majority of our parameter estimates are derived from 30-year surface wind and mixed-layer properties documented by Peralta-Ferriz and Woodgate (2015) (hereafter PFW15). These values are estimated using data collected between $1979-2012$ for five separate Arctic Regions: Chukchi Sea, Southern Beaufort Sea, Canada Basin, Eurasian Basin, and Barents Sea. We therefore provide climatological and trend estimates of $\eta$ and $Fr$ for each of these regions. Details on our choice for each parameter are given below.

We estimate the summer mixed-layer depth ($z_0$) and the buoyancy difference between the mixed layer and deeper ocean ($\Delta B$) using mixed-layer properties reported by PFW15, where we set deeper ocean layer properties equal to observed winter mixed-layer properties. This is similar to PFW15, who also considered a 2-layer ocean associated with summer and winter mixed-layer properties to examine variations in the seasonal pycnocline. Specifically, we use their reported below-ice mixed layer depth and salinity climatological averages in April and July (see their Figures 6 and 8) for the winter and summer mixed layers, respectively, to estimate averages in $z_0$ and $\Delta B$. For our trends in $z_0$ and $\Delta B$, we use their below-ice summer mixed-layer depth and salinity trends that ranged from $-3.9$ m/decade to $+3.3$ m/decade and $-1.9$ psu/decade to $+2.9$ psu/decade, respectively, depending on the region (see their Figure 14). Trends in regions that are reported to be insignificant by PFW15 were taken to have no trend.

We use a climatological pan-Arctic ice speed $U = 0.2\,\mathrm{m\,s^{-1}}$, consistent with measurements of ice speeds collected in both the 1970s (McPhee and Smith, 1976) and the 2000s (Cole et al., 2017). For each Arctic region, we estimate the decadal trend in ice velocity using 30-year regional near-surface wind trends reported by PFW15, and applying the "rule of thumb" introduced by Thorndike and Colony (1982) that the ice speed is approximately $2\%$ of the surface wind speed. With this approach, we estimate ice speed trends as large as $+1.6\,\mathrm{m\,s^{-1}}$/decade, depending on the region (with some regions indicating no trend).

Lastly, lacking data for climatological or decadal trends of ice keel drafts ($h$), we instead repeat our analysis using two different keel sizes. That is, for each region, we compute $\eta$ twice: first setting $h = 7.45$ m, the average keel draft reported by Wadhams (2012) and Kvadsheim (2014), and second using $h = 18.6$ m, which is chosen to examine the same range of $\eta$ that we were able to examine in our simulations.

Fig. 9 shows the resulting climatological averages (numbered boxes) and five- or fifteen-year estimates (solid or dashed arrows, respectively) of $\eta$ and $Fr$ for 10 scenarios: one for each of the five Arctic regions using one of two keel sizes: 7.45 m





(orange) and $18.6$ m (cyan). Five-year estimates were chosen for some regions (Chukchi Sea and Southern Beaufort Sea) because their two-layer stratifications become unstable (summer mixed-layer becomes saltier than winter mixed-layer) after some number of years (*e.g.*, Southern Beaufort Sea after 7.5 years) due to their salinity decadal trends.

On average, we find that each scenario has $0.45 < Fr < 0.76$, where differences in ice keel size explain most of the variation in $\eta$ (as expected by our choice of keel sizes, see above). These ranges roughly double after considering approximately 68% of

the variability (indicated by cyan and orange shadings). Our trends estimates indicate that $Fr$ is increasing for every scenario, with the largest positive trend in the Southern Beaufort Sea ($+0.62$ per decade) and smallest increase in the Barents Sea ($+0.039$ per decade). In contrast, we find both positive and negative trends for $\eta$, depending on the region. We estimate for the $7.45$ m ($18.6$ m) keel that $\eta$ is decreasing at a rate of $0.29$ ($0.71$) per decade in the Southern Beaufort Sea, increasing at a rate of $0.70$ ($1.8$) and $0.032$ ($0.08$) per decade in the Canada and Eurasian Basins, respectively, and does not change in the

Chuckchi and Barents Seas due to insignificant trends in the mixed-layer depth.

Next, we consider the climatological averages together with the results from our simulations (based on where cyan and orange shadings overlap with colored shapes; Fig. 9). For scenarios with large ice keels ($h = 18.6$ m; cyan shadings), we find that all stirring regimes are expected to occur except the Unstable Supercritical upstream regime. These simulations were associated with mixing that could vary by up to 50% upstream and 170% downstream of the background mixing rate (Table

2). By contrast, scenarios with smaller ice keels ($h = 7.45$ m; orange shadings) span a smaller range of Fr-$\eta$ space and are only expected to be associated with the Subcritical upstream stirring regimes and the Lee Wave and Fast-Laminar downstream stirring regimes. These simulations were associated with mixing that only varied up to 10% upstream and 20% downstream of the background mixing rate. Again, we caution against over-interpreting these numbers (see below for more on this point). Finally, the estimated trends in $Fr$ and $\eta$ associated observed decadal changes to ice speeds and upper-ocean properties (arrows)

suggests that all stirring regimes and associated variations in mixing across our entire suite of simulations could be relevant below a changing Arctic ice cover. This suggests that more robust observational estimates, particularly of ice keel sizes and speeds, will be necessary to properly estimate current and future below-ice mixing.

## 5.2 Limitations

We have presented, to our knowledge, the first set of idealized experiments specifically aimed at examining how ice keel size

and speed impacts ocean mixing. While our findings are novel, we emphasize that this work is an initial step at examining this problem, and that more realistic simulations will be necessary to properly parameterize ice keel-driven mixing in coupled models. We expand on the limitations of our current setup below.

First, our simulations use two dimensions to model what is a three-dimensional process in the ocean. Since keels can extend $100$ m in the transverse direction (Topham et al., 1988), we can consider that our simulations are quantitatively representative

of the flow in wide keels' cross-sections far away from their lateral ends. Note that by assuming that there are no ridges or irregularities on the keel, we ignore additional generators of turbulence, and thus of stirring. This might underestimate the mixing that could happen. However, our enhanced viscosity yields a thick viscous boundary layer, which might compensate for the absence of ice roughness.





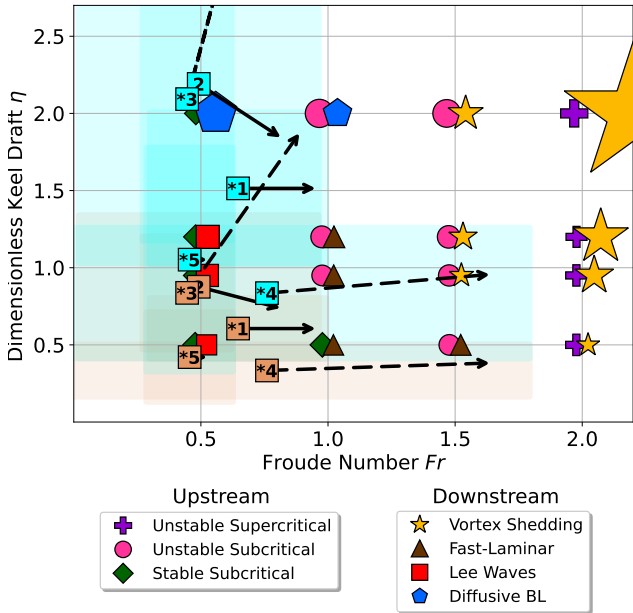

**Figure 9.** Upstream and downstream regime classification for each regime in dimensionless parameter space $(Fr, \eta)$ (Fig. 2) overlaid with range estimates of five Arctic regions: Chukchi Sea (1), Southern Beaufort Sea (2), Canada Basin (3), Eurasian Basin (4), Barents Sea (5). The simulation regime markers are scaled by their respective mixing rates. The orange and cyan square markers indicate climatological average $(Fr, \eta)$ values for the $7.45$ m and $18.6$ m keel, respectively. The transparent boxes indicate one standard deviation range in $Fr$ and $\eta$. The arrow for each marker points to where the average will be in five years (solid) or fifteen years (dashed) based on observed 30-year trends. Note that for the $18.6$ m keel (cyan), the Canada Basin (3) is predicted to reach $(0.87, 4.72)$. Markers with (*) indicate that a mixed-layer or wind speed trend was reportedly insignificant.

Second, our values of viscosity and diffusivity being large in order to ensure numerical stability has additional consequences.
Mainly, they reduce the buoyancy Reynolds number $Re_b$, which can be thought as the ratio of the strength of turbulent insta-
bilities that can induce vertical mixing to the combined strength of stratification and viscosity that impede vertical motions
and mixing. Buoyancy Reynolds number has been shown to control diffusivity and diapycnal mixing rates (Shih et al., 2005;
Mashayek et al., 2017). In particular, for small $Re_b$ (i.e., weak turbulent flows and/or large stratification and viscosity) as in
our simulations, mixing rates and diffusivity tend to be small because the turbulence-induced instabilites are not strong enough
to overcome the stabilizing effects (Bouffard and Boegman, 2013; Salehipour and Peltier, 2015). In other words, our mixing
instead tends to happen through laminar diffusion of buoyancy across smooth density gradients, whereas in a more realistic
fluid with larger $Re_b$, such mixing would happen through episodes of small-scale turbulence in the pycnocline or more widely-
observed turbulent hydraulic jumps downstream (cf. e.g. Winters and Armi, 2012; Lawrence and Armi, 2022). Our upstream
Unstable Subcritical and downstream Vortex Shedding regimes feature such mixing events, which we may have observed more
generally had we been able to use a finer resolution and smaller values of $\mu$. We leave it to future studies to investigate how
smaller viscosity and enhanced turbulence translate into qualitative differences for $\overline{\Phi}_U$, $\overline{\Phi}_D$, $\overline{Z}_U$ and $\overline{Z}_D$.



Third, because our simulations are two-dimensional, the turbulence we see sometimes follows an inverse cascade of energy, especially for vortices entirely contained in either the mixed or the deeper ocean layer. That is, instead of twisting and stretching into smaller and smaller structures, vortices of the same sign merge and become larger and larger in diameter (Vallis, 2017). This process could contribute to artificially expanding the diameters of the vortices we see in the upstream Unstable Sub- and Supercritical regimes, and the downstream Vortex Shedding regime, which cause the largest values of $\overline{\Phi}_U$ and $\overline{\Phi}_D$ as well as the deepest mixing depths $\overline{Z}_U$ and $\overline{Z}_D$. As a first consequence, we might be overestimating some values for our mixing depth. As for $\overline{\Phi}_U$ and $\overline{\Phi}_D$, the inverse cascade may have two separate effects. First, the large size of the vortices in the mixed layer might lead to more entrainment of deep fluid into the mixed layer, enhancing local density gradients and therefore turbulent mixing. On the other hand, mixing in stratified flows occurs through a host of secondary instabilities that are three-dimensional in nature (*e.g.*, Mashayek and Peltier, 2012a, b). Such instabilities are absent from our two-dimensional simulations, which is likely to have qualitative and quantitative consequences on the values of effective diffusivities.

Fourth, in the Arctic Ocean, there are roughly 2.0 to 4.5 keels per kilometer (Wadhams, 1981), which roughly translates to a keel every $28z_0$ to $63z_0$. Although we do not account for additional keels in our domain, any fluid phenomena such as vortices, solitary waves, and lee waves develop within $25z_0$ of the keel. This means that our flow regimes are very localized to the keel and have enough space to realistically form in the Arctic Ocean. However, the establishment of these flow regimes assumes an upstream and downstream flow similar to the initial conditions that are not affected by surrounding keels. An upstream keel would mix (to varying degrees) the established pycnocline, making the incident flow more smoothly stratified, thus changing the dynamics of the situation. On the downstream side of the domain, if the flow is in the Subcritical Unstable regime, a downstream keel could generate upstream-propagating internal waves and would further complicate the problem.

With these considerations in mind, we caution the reader against directly comparing our numbers and simulated processes with measurements made in the Arctic Ocean. However, comparing the same metrics across the simulations, the regimes and parameters we defined should provide useful qualitative information about which real-world situation might lead to more or less mixing than another.

# 6 Summary

We performed sixteen numerical simulations of two-layer fluid flow over the cross-section of an ice keel (Figure 1), targeted at understanding how mixing varies for the bulk Froude number $Fr$ and relative keel draft $\eta$. We estimated that the Arctic Ocean should exhibit a range of non-dimensional ice keel speeds of $0 < Fr < 2.5$ and non-dimensional ice keel depths of $0 < \eta < 5.6$. We were able to explore approximately half of this parameter space ($0 < Fr < 2.0$ and $0 < \eta < 2.0$) and found within this domain a fairly broad range of dynamics. Specifically, we characterized the variety of flows that were generated from flow across the ice keel using three upstream and four downstream stirring regimes (Figures 2, 3, and 6). Further, we found that these dynamics were essential for explaining differences in mixing across simulations (Figures 7-8).

What do these results mean for accurately representing ice keel mixing below a rapidly changing sea ice cover in coupled models? The range of mixing levels across simulations further supports previous studies that found that ice-ocean momentum



exchange is sensitive to factors aside from wind speed. The majority of these studies suggested that changing ice keel sizes must be considered because larger ice keels can increase ice-ocean drag. Our results have added an additional level of complexity to this paradigm by demonstrating that, in contrast to ice-ocean drag, ocean mixing does not increase monotonically with larger keel depth and speed. Instead, our simulations demonstrate that larger, faster ice keels can substantially inhibit or enhance ocean mixing, and that this depends on the generation and propagation of vortices and turbulence. Finally, our results highlight that

improved estimates of current and future below-ice mixing will require two key pieces of information: (1) better observational estimates of ice keel characteristics and (2) more realistic numerical simulations aimed at examining how the size and speed of ice keels impact ocean mixing.

*Code availability.*  The simulation and processing code as well as the calculations used to create Fig. 9 can be found at https://doi.org/10.5281/zenodo.8170312.

*Video supplement.*  Videos of all sixteen simulations can be found at https://doi.org/10.5281/zenodo.8169338.

*Author contributions.*  RMC developed the simulation code with guidance from MGS. SDA made significant modifications to the simulation code. VEZ and SDA created the simulation post-processing code for the mixing analysis. SDA ran the simulations with the post-analysis performed by everyone. The paper was primarily written by SDA, ER and NG with the figures created by SDA and RMC. Everyone edited and revised the paper for submission.

*Competing interests.*  No competing interests are present.

*Acknowledgements.*  S.D.A., R.M.C., M.G.S. and N.G. acknowledge the support of the Natural Sciences and Engineering Research Council of Canada (NSERC) [funding reference numbers RGPIN-2015-03684 and RGPIN-2022-04560]. S.D.A was supported by ArcticNet, a Network of Centres of Excellence Canada (project 52551). V.E.Z. acknowledges the support of National Science Foundation OCE1756752 and OCE2220439. E.R. was supported by the National Sciences and Engineering Research Council of Canada (NSERC) PDF award and

the NSERC Canada-150 Chair (Award G00321321). E.R. is grateful to the researchers, staff and students of the Centre for Earth Observation Science for support received during preparation of this manuscript. Computations were performed on the Niagara supercomputer at the SciNet HPC Consortium (Loken et al., 2010; Ponce et al., 2019). SciNet is funded by: the Canada Foundation for Innovation; the Government of Ontario; Ontario Research Fund - Research Excellence; and the University of Toronto.





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
