# Peer review of "Two-dimensional Numerical Simulations of Mixing under Ice Keels"

_EGUsphere, 2023_

## Referee Comment (RC2)

The paper is focused on the investigation of irreversible (diabatic) mixing of upper ocean layer by ice ridge keels. Effect of diabatic mixing is related to salt diffusion in conditions of complicated motion of stratified sea water. Numerical simulations with created in a spectral partial differential equation solver Dedalus were used for the investigations. Although the calculation time is few tens of minutes, the effect has long-term consequences and could be applied to solve climate problems.

There are following comments to the paper:

1. Nonlinear terms in momentum balance equations (1) and (2) are different from standard expressions $u\nabla u$. Viscous terms $\nu\partial q/\partial z$ and $-\nu\partial q/\partial x$ in equations (1) and (2) are also different from standard form $\nu\Delta u$. Equations (1) and (2) are different from the momentum balance equations considered in the papers of Skyllingstad et al (2003) and Hester et al (2021) given in the reference list. More detailed explanation of equations (1) and (2) is necessary for improving of understanding of the problem statin.

2. Diabatic mixing is caused by salt diffusion in conditions of internal waves excited by the interaction of the ice keel with water flow leading to adiabatic stirring. Coefficient of salt diffusion is set to $\mu = 2 \cdot 10^{-3}$ m²/s in numerical simulations. This value is much larger the molecular salt diffusion $\sim 10^{-9}$ m²/s. The large value of $\mu$ is chosen to dissipate eddies smaller than the resolution of the grid (line 96). Further increasing $\mu$ influence diabatic mixing according to formula (12). Please give more physical reasons for the choice of numerical value of $\mu$.

3. Kinematic viscosity $\nu = 2 \cdot 10^{-3}$ m²/s is also larger molecular kinematic viscosity of $1.78 \cdot 10^{-6}$ m²/s. Is it turbulent eddy viscosity? Please explain physical sense of $\nu$.

4. Authors ignore thermal effects assuming water temperature equals -2 C. The water temperature is assumed depending on salinity (lines 91-92). Temperature at ice-water interface should be equal to the freezing point, and outside of the interface temperature is equal the freezing point or higher. Adiabatic mixing and diabatic stirring lead to increasing of water salinity and decreasing of the freezing point at ice-water interface. Decreasing of the freezing point influences ice melt leading to decreasing of water salinity and density near the interface. How strong this effect is in long term perspective?

5. Estimates of ice drift speed using wind drag coefficient are not correct in the Barents Sea regions with relatively strong semidiurnal tide and influence of Spitsbergen, Franz Josef Land and Novaya Zemlya. Semidiurnal tide is stronger in the Barents Sea than in East Arctic regions. Speed of semidiurnal tidal current may exceed 1 m/s in the region between Bear and Hopen Islands. Also, water temperature below drift ice is frequently higher than -2 C in the Barents Sea. Depending on tidal phase and wind it varies from -1C to -1.9C.

6. All ice ridges in the Barents Sea are the first-year ridges. Shape of their keels is not no smooth as it is considered in the papers. Ridge keels are not completely consolidated, and macro porosity of their unconsolidated parts vary in the range

20-40%. Water can penetrate inside ridge keels, and boundary condition with zero normal velocity should be modified.

I recommend major revision of the paper to improve model description and thermodynamic justification of applicability of obtained results for the investigation of long-term processes in upper ocean layer.

---

## Author Comment (AC1)

**Two-Dimensional Numerical Simulations of Mixing under Ice Keels**

**Response to Reviewer #1**

We would like to thank the reviewer for their thoughtful and detailed evaluation of our manuscript. We believe the manuscript has benefited greatly from your suggestions. Below, each question/suggestion from the review is listed followed by our response to it in blue. In suggestions where content was added or modified to the manuscript, the content is listed below the response *italicized*.

**Reviewer comments:**

As a reader and editor for Ocean Science, I found the topic and content of this submission to be more suitable for Ocean Science. If you agree, the submission could be transferred from TC to Ocean Science and retain the reviews and discussion.

- We thank the reviewer for this suggestion. After careful consideration, we still believe that TC is a better fit overall. While the mixing aspect may be more suited towards OS, our original goal was to make the paper and its analysis accessible to TC in hopes of extending this topic further observationally.

Li 146: definition of the buoyancy difference, DeltaB. There is some inconsistency with using the summer and winter DeltaS bounds and the mixed layer depth from Peralta-Ferriz and Woodgate. They used a density step threshold of 0.1 kg/m3 which roughly translates to a buoyancy difference of 1e-3 m/s2. But your range is from about three times to 75 times that value. I certainly do not ask for new simulations. However, you should discuss the implications of this.

- Thank you for pointing this out; we have clarified this potential source of confusion in several ways:
  - We clarify that our approach is consistent with the 2-layer model presented in PFW (see their Figure 9c).
  - We note that the threshold of 0.1 kg/m^3 is used to estimate mixed-layer depths over a variety of seasons and regions. This value provides an estimate of the minimum density change necessary to indicate the base of the mixed layer. Given that the largest surface stratification occurs in summer, we expect our value to indeed be larger than this threshold. That being said, a two-layer model will, by construction, tend to yield a larger than observed stratification (though PFW note finding many "step-like" summer profiles). We clarify this in our methods, where we compute the buoyancy difference (Delta B):
    - *Note that this will yield larger Delta B-values than those resulting from using the 0.1 kg/m^3 density step from PFW, from which we obtained our mixed-layer depth. The discrepancy in Delta B results from our*

> *choice to define the densities rho_1 and rho_2 using summer and winter values, where -- by contrast -- PFW defines rho_2 using a value representing the transition between the summer and winter layers. We believe that the former better encapsulates summer conditions reminiscent of PFW's two-layer model (see their Figure 9c).*
> - We now note that this could impact our results by inhibiting the impact of mixing due to a stronger stratification in our Limitations section:
>   - *First, we choose large values of viscosity and diffusivity to ensure numerical stability, but these choices have further consequences in addition to a strong stratification from our two-layer model. Mainly, they reduce the buoyancy Reynolds number Re_b, which can be thought of...*

Referring to Figure 8, if the vertical reach of mixing is roughly two times z0, i.e., one additional mixed layer depth below the mixed layer depth of z0, for a relatively thick pycnocline layer (in real ocean) below a shallow mixed layer (say, z0=10 m and the diffuse pycnocline thickness is 20 m), mixing will not penetrate below the pycnocline and will not contribute to entrainment into the mixed layer. I would like to see some discussion about this.

- Thank you for pointing this out. Our pycnocline is approximately 0.5z_0 after settling, which is on the thinner side for seasonal pycnoclines. Indeed, regimes that have large mixing rates due to entrainment of the deeper ocean (e.g., Vortex Shedding regime) would likely see reduced mixing rates for a thicker pycnocline because, as you mention, they would entrain less deep water into the mixed layer. We have added a comment that brings this to the reader's attention. Note that a thicker pycnocline might make it easier to entrain partially-mixed water (with density between the summer and winter layer densities) due to a weaker buoyancy frequency, but it is uncertain how much this would compensate for the decrease in mixing rates mentioned above. We have added a couple sentences summarizing this at the end of our results:
  - *In addition, our pycnocline is on the thinner side (~0.5z_0 after settling) in comparison to other seasonal pycnocline measurements (thickness values beyond z_0 are possible, as seen in PFW). Simulations with large mixing rates due to entrainment of the deeper ocean ($\ZU$ or $\ZD\geq1$, as in the Vortex Shedding regime) would likely see reduced rates for a thicker pycnocline.*

Discussion includes "Implications" (actually, climatological and trend estimates), and "Limitations". I would like to see some discussion of the results too, on the findings in general but also including perhaps a discussion on the context/applicability of other studies on flow over sills etc, on the excluded interfacial/internal wave drag and related processes.

- We appreciate this suggestion but were also concerned that adding additional sections would seem repetitive. After careful consideration, we ultimately chose not to include more discussion of the results or the similarity of our work to studies of flow over sills beyond what was presented in Section 4 and Section 1, respectively.

Opening paragraph: the narrative suggests the issue is a misrepresentation of ocean mixing under ice-covered waters. But this is only part of the story of the poor performance of large-scale models.

- We agree and thank the reviewer for this suggestion. We have addressed this point by "softening" the language through the first paragraph to more accurately reflect \*possible\* implications of improving our understanding of ice-ocean interactions, as follows:
    - *... potentially contributing to an unrealistic representation of the Arctic halocline…*
    - *This may have direct implications for biases in simulated circulations of Pacific and Atlantic Water and possibly sea ice retreat…*

Second paragraph: studies diverge on the effect of decreased sea ice cover on potentially increasing wind-induced mixing. The literature review on this is not up-to-date. There are several studies that attempted to quantify the change in the near-inertial energy field in the Arctic in recent decades and how this is influenced by the sea ice cover.

- We thank the reviewer for this suggestion, which we have addressed by altering our second paragraph to emphasize that this is still an open area of research, and have included more up-to-date publications:
    - *The shrinkage of this "sea ice lid" has allowed the wind to interact directly with the ocean, increasing wind-driven momentum transfer into the ocean but yielding uncertain effects for vertical mixing (Guthrie et al., 2020; Lincoln et al., 2016; Dosser et al., 2021; Fine & Cole, 2022).*

Third paragraph: I am not a sea-ice expert, but I suspect the cited literature on changes in sea ice thickness and age may be outdated (newest 2018). Given that this is a submission to TC, the state-of-the-art can be improved.

- We have added three papers (Sumata et al., 2023; Zhang et al., 2021; Meier & Stroeve, 2022) published in the last three years to our cited literature in paragraph three. These studies come to similar conclusions as were reached in the older literature, but on a larger and more recent data set. Thank you for pointing this out.

Li 46-48: Agreed, but please also include some seminal papers from McPhee on the effects of under-ice roughness. (Actually, the only McPhee reference cited is from 1976.)

- Thank you for noting our lack of reference to McPhee's work in ice ocean boundary layers. We have reviewed McPhee's publications and updated this section to reference three additional, relevant papers, concerning vertical mixing (McPhee, 1983), internal wave generation (McPhee & Kantha, 1989), and ice-ocean drag (McPhee, 2012).

Li 51-52: Although not directly an ice-keel study, laboratory experiments in cases where the ice floe protrudes into the pycnocline reported in Carr et al (2019) can also be insightful. [Carr, M., et al. (2019). Laboratory experiments on internal solitary waves in ice-covered waters. Geophysical Research Letters, 46, https://doi.org/10.1029/2019GL084710]

- Thank you for bringing this paper to our attention. We have added this study to our literature review on mixing underneath ice floes in the introduction (fifth paragraph):
  - *Carr et al. (2019) and Zhang et al. (2022) ran numerical experiments of internal solitary waves (non-linear, non-hydrostatic oscillations of the pycnocline) impinging on floe edges and ice keels, respectively, reporting that this interaction can result in the creation of secondary waves and turbulence.*

Li 67: one of three and one of four stirring regimes can be confusing for the reader. Perhaps simply "we categorize the different stirring regimes in the upstream and downstream of the keel for each simulation."
- We agree and have implemented the suggested change.

Li 84: I generally agree to ignore Coriolis in this study, but note that you do not need to go far from the boundary layer before the effect of rotation has a significant influence on the mixing length (so-called outer layer, see the McPhee book or book chapters).
- Thank you for bringing this to our attention. Using a bulk stress estimate for the friction velocity ($u\_*$) for a variety of our keel speeds and McPhee's formulation for the surface layer extent ($z\_sl=0.05*u\_*/f$), we find that the surface layer can extend from a couple meters to 18m in our simulations (for a conservative drag coefficient of $5.5*10^\wedge(-3)$ considering a keel is present); we note that the large surface-layer extent is a byproduct of our large keel speeds. The majority of the mixing we observe occurs within this surface layer, where rotation does not influence the mixing length. Additionally, we believe rotational effects would not be observed in the outer layer if we modified our model to account for rotation, as our simulation timescales are prohibitively short (shorter than an inertial period). We have added a few sentences elaborating on this in our Methods section, where we remark that we neglect Coriolis terms:
  - *Note that one does not need to go far from the ice boundary (a couple meters for typical surface stresses) before rotation can influence the mixing length (McPhee, 2012); however, our timescales are sufficiently short (shorter than an inertial period) to neglect rotational effects.*

Fig 1 caption can also define phi, sigma and h or refer to text. Throughout, please use Roman Fr for the Froude number and Re for the Reynolds number.
- We thank the reviewer for these suggestions and we've implemented the suggested changes both in the Figure 1 caption and throughout the text.

Li 169: I'm not sure how to interpret this Re when the viscosity is replaced with a large value that mimics turbulent viscosity. I guess it is common practice in modeling. One of its implications, in mixing through low buoyancy Re is discussed later. Perhaps here a comment is also needed, about this implication and others if any, for the non-modeler reader.
- Yes, it is common practice in the modeling community and borne out of necessity, because numerical grids can accommodate fine velocity gradients only up to a point. We thank you for reminding us that readers of The Cryosphere may not be familiar

with this concept, and have added a sentence to clarify where we introduce our diffusivities:

- *We choose such large values for nu and mu to dissipate eddies smaller than the resolution of the grid, similar to the choice of Zhang et al. (2022), which keeps our numerical cost tractable. These values may have quantitative consequences on mixing but should preserve our qualitative conclusions, as we discuss in Section 5.2.*

Eq.10: Why is the sorted density not a function of the horizontal distance, x?

- The sorted density field, or the background density field, cannot vary in the x direction. If it did, then it would necessarily contain available potential energy. That is, if there were horizontal gradients in the sorted density profile, it would no longer be describing the minimum potential energy state because globally over the entire domain, the lighter parcels will rise and denser parcels will sink due to the gravitational force. By definition, however, the background density field corresponds to a state of minimum potential energy of the system, and requires an arbitrary fluid parcel to be located at a strictly lesser depth than every denser parcel. This background density field is a purely mathematical construct and provides a useful reference point against which to measure the location of the global center of fluid mass. For the fluid system as a whole, irreversible mixing changes this global center of mass (by converting available potential energy into background potential energy). In simulations, because we have a limited domain and because we know the density distribution throughout the domain at every time step, we can compute this change in the global center of mass at every time step by finding the minimum potential energy state of the system at that time. This is different from common methods of estimating irreversible mixing in observational work that typically use localized vertical temperature gradients (e.g., Osborne-Cox models). Instead, as described in the manuscript, the method we use here follows Winters et al (1995) to decompose potential energy into the background and available components to diagnose mixing. We appreciate the reviewer's comment, but after careful consideration, we decided that an in-depth explanation of the energy decomposition beyond what is already included in the paper and references to Winters et al (1995) would distract from the main narrative of the study.

Li220: because of the division by [the molecular diffusivity] mu, …(to help the reader)

- We agree with your suggestion and have implemented this change.

Li 257: cross-reference should be to section 2.1

- Yes, thank you; we have implemented this change.

Li 271: please clarify "ahead" of the keel, by using upstream or downstream

- Thank you for this comment. We have replaced it with "upstream".

Fig 5 caption: the regime was defined without "Waves" in it [Unstable Subcritical regime]

- Thank you for catching this error.

Li 272: Fig3b shows the streamline not the vorticity. Perhaps use : "as we can see in the streamlines in Fig 3b… and in the spanwise vorticity field in Fig 5a.
- We agree and have reworded this line:
  - *…as evidenced by the streamlines in Figure 3b and by the spanwise vorticity field in Figure 5b, in the region $40<x/z\_0<70$.*

Li 295: please comment on the presence or lack of mixing for this regime
- We intended this section to be a kinematic description of the regimes, with discussions of mixing reserved for Section 4. In Section 4 (Li 374), we briefly discuss the lack of mixing in the Fast-Laminar regime due to its predominantly flat isopycnals and very infrequent vortex advection. We believe that this sufficiently explains our reported mixing values.

Table 2 Caption: Missing "mixing" before depths. A missing closing bracket in the end.
- Thank you for catching this error; we have revised the caption.

Li 323: could insert: "… the largest mixing rate [in the upstream] does not …"
- We agree and have reworded this line.

Fig 8 caption: could also mention overbar(Z) = 1 equals the mixed layer depth, z0.
- We agree and have implemented this change:
  - *As a reminder, $\overline{Z}_{\Omega}=1$ implies a mixing depth equal to the mixed-layer depth.*

Li 409: using a constant speed is an over simplification that is worth commenting
- Thank you for bringing this up, especially since this oversimplification may be the biggest in Section 5.1. We have added a comment noting that this is a simplification when we introduce the climatological pan-Arctic ice speed U:
  - *This is a vast simplification, as in reality U varies largely across the Arctic due to spatial and temporal variability in wind forcing and semidiurnal tides.*

Li 413: typo in the ice speed trend. should be cm/s?
- Thank you for catching this error. Upon further checking, we also noted that the largest ice-speed trend was 3.2cm/s per decade and not 1.6cm/s per decade.

Li 478-485: On the positive side, your inferences can actually be representative of a floe. Your upstream and downstream control volumes are roughly (30-40)z0 long. For a 10 m MLD, this is roughly 300 m. One keel every 300 m should be typical (as you mention with reference to Wadhams). So effectively, your mixing calculations could be representative of the floe and not as local as you imply here.
- This is a good point and we agree to a large extent. For medium-sized floes with one keel, our analysis should be fairly representative of the dynamics away from the floe

edges. The edges can create regions of flow separation, which can lead to entrained vortices and mixing (see Hester et al. (2021)). For a sufficiently long floe with a keel away from the edges, we anticipate that edge effects would not significantly impact the mixing we observe around the keel. We have added a comment summarizing the above discussion:

- *...and could be representative of the dynamics underneath an entire floe with one keel (ignoring the floe's edges).*

**References**

(PFW) Peralta-Ferriz, C., & Woodgate, R. A. (2015). Seasonal and interannual variability of pan-Arctic surface mixed layer properties from 1979 to 2012 from hydrographic data, and the dominance of stratification for multiyear mixed layer depth shoaling. Progress in Oceanography, 134, 19–53. https://doi.org/10.1016/j.pocean.2014.12.005

Guthrie, J. D., & Morison, J. H. (2021). Not Just Sea Ice: Other Factors Important to Near‑inertial Wave Generation in the Arctic Ocean. Geophysical Research Letters, 48(3). https://doi.org/10.1029/2020GL090508

Lincoln, B. J., Rippeth, T. P., Lenn, Y., Timmermans, M. L., Williams, W. J., & Bacon, S. (2016). Wind‑driven mixing at intermediate depths in an ice‑free Arctic Ocean. Geophysical Research Letters, 43(18), 9749–9756. https://doi.org/10.1002/2016GL070454

Dosser, H. v., Chanona, M., Waterman, S., Shibley, N. C., & Timmermans, M. ‑L. (2021). Changes in Internal Wave‑Driven Mixing Across the Arctic Ocean: Finescale Estimates From an 18‑Year Pan‑Arctic Record. Geophysical Research Letters, 48(8). https://doi.org/10.1029/2020GL091747

Fine, E. C., & Cole, S. T. (2022). Decadal Observations of Internal Wave Energy, Shear, and Mixing in the Western Arctic Ocean. Journal of Geophysical Research: Oceans, 127(5). https://doi.org/10.1029/2021jc018056

Sumata, H., de Steur, L., Divine, D. v., Granskog, M. A., & Gerland, S. (2023). Regime shift in Arctic Ocean sea ice thickness. Nature, 615(7952), 443–449. https://doi.org/10.1038/s41586-022-05686-x

Zhang, F., Pang, X., Lei, R., Zhai, M., Zhao, X., & Cai, Q. (2022). Arctic sea ice motion change and response to atmospheric forcing between 1979 and 2019. International Journal of Climatology, 42(3), 1854–1876. https://doi.org/10.1002/joc.7340

Meier, W., & Stroeve, J. (2022). An Updated Assessment of the Changing Arctic Sea Ice Cover. Oceanography, 35, 10–19. https://www.jstor.org/stable/27182690

McPhee, M. G. (1983). Turbulent heat and momentum transfer in the oceanic boundary layer under melting pack ice. Journal of Geophysical Research: Oceans, 88(C5), 2827–2835. https://doi.org/10.1029/JC088iC05p02827

McPhee, M. G., & Kantha, L. H. (1989). Generation of internal waves by sea ice. Journal of Geophysical Research: Oceans, 94(C3), 3287–3302. https://doi.org/10.1029/JC094iC03p03287

McPhee, M. G. (2012). Advances in understanding ice – ocean stress during and since AIDJEX. Cold Regions Science and Technology, 76–77, 24–36. https://doi.org/10.1016/j.coldregions.2011.05.001

Winters, K. B., Lombard, P. N., Riley, J. J., & D'Asaro, E. A. (1995). Available potential energy and mixing in density-stratified fluids. Journal of Fluid Mechanics, 289, 115. https://doi.org/10.1017/S002211209500125X

Hester, E. W., McConnochie, C. D., Cenedese, C., Couston, L.-A., & Vasil, G. (2021). Aspect ratio affects iceberg melting. Physical Review Fluids, 6(2), 023802. https://doi.org/10.1103/PhysRevFluids.6.023802

---

## Author Comment (AC2)

**Two-Dimensional Numerical Simulations of Mixing under Ice Keels**

**Response to Reviewer #2**

We would like to thank the reviewer for their thoughtful and detailed evaluation of our manuscript. We believe the manuscript has benefited greatly from your suggestions. Below, each question/suggestion from the review is listed followed by our response to it in blue. In suggestions where content was added or modified to the manuscript, the content is listed below the response *italicized*.

**Reviewer comments:**

Nonlinear terms in momentum balance equations (1) and (2) are different from standard expressions .Viscous terms  and  in equations (1) and (2) are also different from standard form. Equations (1) and (2) are different from the momentum balance equations considered in the papers of Skyllingstad et al (2003) and Hester et al (2021) given in the reference list. More detailed explanation of equations (1) and (2) is necessary for improving of understanding of the problem statin.

- The equations presented in our paper and in Hester et al. (2021) are equivalent, but presented in a different form. They derive from the vector identity $(\vec{v} \cdot \vec{\nabla})\, \vec{v} = \vec{\omega} \times \vec{v} + \vec{\nabla}(|v|^2/2)$, with $\vec{\omega} = \vec{\nabla} \times \vec{v}$ the vorticity. The left-hand side of the first equation is the more familiar version you are referring to, and the right-hand side is the formulation that Hester et al. (2021) chose for their implementation. In this formulation, the "pressure gradient" is the gradient of the familiar thermodynamic pressure, plus that of $|v|^2/2$. This minor re-casting of the advection term aside, the velocity and buoyancy solutions are strictly the same. We have summarized the above in our methods:
    - *The equations are written in a computationally advantageous form by decomposing the advection term into the sum of the Lamb vector (-wq, uq) and a gradient term incorporated into the pressure gradient.*

Diabatic mixing is caused by salt diffusion in conditions of internal waves excited by the interaction of the ice keel with water flow leading to adiabatic stirring. Coefficient of salt diffusion is set to m2/s in numerical simulations. This value is much larger the molecular salt diffusion  m2/s. The large value of  is chosen to dissipate eddies smaller than the resolution of the grid (line 96). Further increasing influence diabatic mixing according to formula (12). Please give more physical reasons for the choice of numerical value of.

- The choice of our salt and momentum diffusivities originate completely from numerical stability. That is, in no way could we fully resolve the large separation of scales between the scales of molecular diffusion (mm) to the domain of interest (10-100m) within a reasonable computational time and with reasonable computational cost. So, it is a common practice in modeling to increase the value of viscosity and

diffusivity in order to "shrink" the range of scales to something that we could fully resolve in a reasonable time. We thank the reviewer for pointing this out and have added a comment in our methods that summarizes this rationale when introducing our diffusivities:

- *We choose such large values for nu and mu to dissipate eddies smaller than the resolution of the grid, similar to the choice of (Zhang et al., 2022), which keeps our numerical cost tractable. These values may have quantitative consequences on mixing but should preserve our qualitative conclusions, as we discuss in Section 5.2.*

Kinematic viscosity m2/s is also larger molecular kinematic viscosity of m2/s. Is it turbulent eddy viscosity? Please explain physical sense of.

- Please see our previous response regarding large diffusivities.

Authors ignore thermal effects assuming water temperature equals -2 C. The water temperature is assumed depending on salinity (lines 91-92). Temperature at ice-water interface should be equal to the freezing point, and outside of the interface temperature is equal the freezing point or higher. Adiabatic mixing and diabatic stirring lead to increasing of water salinity and decreasing of the freezing point at ice-water interface. Decreasing of the freezing point influences ice melt leading to decreasing of water salinity and density near the interface. How strong this effect is in long term perspective?

- Thank you for bringing this to our attention. Yes, by ignoring the melting of the keel we ignore the stabilizing buoyancy flux from the meltwater, which can hinder turbulence and reduce mixing. It should be noted that the rate of melting depends on the keel's ability to stir or mix away the fresh meltwater and pull up heat fluxes from below. That is, the relationship between mixing and melting is one of negative feedback. This complicates the situation and doesn't allow us to qualitatively determine the effect that melting would have in our simulations/regimes in the long term without future study. If our keel speed were physically variable, then melting could reduce drag and allow it to travel faster/further (McPhee, 1983; McPhee, 2012), which is significant in the long term. We have added a paragraph in Section 5.2 explaining the limiting effects of our constant-temperature domains:
  - *Fifth, fixing the temperature of our domain at the freezing point of seawater necessarily suppresses melting of the keel. Aside from structurally changing the keel, melting would produce a stabilizing buoyancy flux of freshwater immediately below the ice that could hinder turbulence and, consequently, mixing. The rate of melting responds to the keel's ability to draw up heat fluxes from below and to mix or stir away the fresh meltwater immediately below the ice (Skyllingstad et al., 2003). For instance, regimes like Vortex Shedding would likely see high melting rates because of their large mixing rates and mixing depths; however, the effects of the stabilizing buoyancy flux from the subsequent meltwater on the regime's mixing rates are uncertain and require further work. If our ice keel speed were physically variable (i.e., influenced by drag), then melting could hydrodynamically ``decouple'' the ice floe and its*

*keel(s) from the upper ocean boundary layer, reducing drag and allowing the floe to travel faster; however, this is beyond the scope of this paper. The reader is referred to McPhee (2012) for more information.*

Estimates of ice drift speed using wind drag coefficient are not correct in the Barents Sea regions with relatively strong semidiurnal tide and influence of Spitsbergen, Franz Josef Land and Novaya Zemlya. Semidiurnal tide is stronger in the Barents Sea than in East Arctic regions. Speed of semidiurnal tidal current may exceed 1 m/s in the region between Bear and Hopen Islands. Also, water temperature below drift ice is frequently higher than -2 C in the Barents Sea. Depending on tidal phase and wind it varies from -1C to -1.9C.

- Thank you for bringing this to our attention. It should be noted that the wind drag coefficient is only used for estimating the ice speed trend based on the wind speed trend. To factor in decadal changes in semidiurnal tides would make this side of the analysis too detailed when we made other, more consequential approximations (e.g., assuming a two-layer density model). Regarding the choice of a constant ice speed across the Arctic, we agree that this may be our largest simplification and that semidiurnal tides could alter this value depending on location. We have added a comment to bring this to the readers' attention:
  - *This is a vast simplification, as in reality $U$ varies largely across the Arctic due to spatial and temporal variability in wind forcing and semidiurnal tides.*

All ice ridges in the Barents Sea are the first-year ridges. Shape of their keels is not no smooth as it is considered in the papers. Ridge keels are not completely consolidated, and macro porosity of their unconsolidated parts vary in the range 20-40%. Water can penetrate inside ridge keels, and boundary condition with zero normal velocity should be modified.

- Thank you for pointing this out. We agree that our boundary condition isn't entirely representative of all ice keels, especially first-year ridges. Flow through porous mediums is a complicated topic and proper treatment would require a more intricate model, which is beyond the scope of the paper. We believe that your first point about the smoothness of our keels is in fact a larger simplification than our boundary condition. Edges or irregularities in the keel can become significant turbulence generators and effectively change the entire flow behavior and hence mixing. To factor in all various forms and irregularities of a keel would require a more statistical approach and a more sophisticated numerical solving scheme, which is left for future work. We have added a couple of sentences discussing this in our limitations section with an emphasis on future work:
  - *In addition, ice keels are conglomerates of ice rubble with varying degrees of porosity. As such, our no-slip condition at the keel boundary is not necessarily realistic, particularly for young keels. Accounting for porous flow would require a more intricate model, which is left for future work.*
  - *Note that, by assuming that there are no ridges or irregularities on the keel, we ignore additional generators of small-scale turbulence, and thus of stirring. This may result in underestimating mixing.*

**References**

Hester, E. W., McConnochie, C. D., Cenedese, C., Couston, L.-A., & Vasil, G. (2021). Aspect ratio affects iceberg melting. Physical Review Fluids, 6(2), 023802. https://doi.org/10.1103/PhysRevFluids.6.023802

Skyllingstad, E. D., Paulson, C. A., Pegau, W. S., McPhee, M. G., & Stanton, T. (2003). Effects of keels on ice bottom turbulence exchange. Journal of Geophysical Research: Oceans, 108(12), 1–16. https://doi.org/10.1029/2002jc001488

McPhee, M. G. (1983). Turbulent heat and momentum transfer in the oceanic boundary layer under melting pack ice. Journal of Geophysical Research: Oceans, 88(C5), 2827–2835. https://doi.org/10.1029/JC088iC05p02827

McPhee, M. G. (2012). Advances in understanding ice – ocean stress during and since AIDJEX. Cold Regions Science and Technology, 76–77, 24–36. https://doi.org/10.1016/j.coldregions.2011.05.001

---

## Referee Report (RR1)

Review of the paper "Two-dimensional numerical simulations of mixing under ice keels" by Sam De Abreu et al.

The paper presents the results of numerical modelling to study the effects of sea water mixing by the keels of drifting ice. The authors considered 16 combinations of the Froude number and keel draft to describe different scenarios of water movement near the keels. The combinations represent typical sizes of ice ridges and speeds of sea currents under Arctic ice. The authors compared the mixing and stirring effects obtained from the simulations and demonstrated, in contrast to ice-ocean drag, that ocean mixing does not increase monotonically with larger keel depth and speed.

The conclusion about increasing ice-ocean drag was made based on previous studies. It is unclear whether this is true for numerical simulations described in the paper or not. It would be useful to estimate form and skin drags on ice keels for different combinations of the Froude number and keel draft based on simulation results.

From Table 2 it can be seen that diapycnal diffusivities are of the order of the kinematic viscosity and salt-mass diffusivity or even larger. The kinematic viscosity and salt-mass diffusivity set to 0.002 m2/s are much smaller the molecular kinematic viscosity (~1.8 10^-6 m2/s) and salt diffusivity (~10^-9 m2/s). Large values of these parameters may be reasonable for numerical simulations. However, a comparison with results from other studies of sea water mixing in the Arctic would be useful for the paper. For example, Liang and Losh (JGR, 2018) write on excessively strong vertical mixing of 1.25x10^-4 m2/s with the background diffusivity coefficient of 5.44x10^-7 m2/s. Fer (Atm. And Oceanic. Science Letters, 2009) reported about mixing coefficients of 10^-6-10^-5 m2/s estimated from the field observations in the Arctic.

I think that the paper needs to be elaborated according to two comments above.